# Major agricultural changes required to mitigate phosphorus losses under climate change

M.C. Ockenden [1], M.J. Hollaway [1], K.J. Beven[1], A.L. Collins[2], R. Evans[3], P.D. Falloon [4], K.J. Forber[1], K.M. Hiscock[5], R. Kahana[4], C.J.A. Macleod[6], W. Tych [1], M.L. Villamizar[7], C. Wearing [1], P.J.A. Withers[8], J.G. Zhou[9], P.A. Barker[1], S. Burke[10], J.E. Freer[11], P.J. Johnes[11], M.A. Snell[1], B.W.J. Surridge [1] & P.M. Haygarth [1]

Phosphorus losses from land to water will be impacted by climate change and land management for food production, with detrimental impacts on aquatic ecosystems. Here we use a unique combination of methods to evaluate the impact of projected climate change on future phosphorus transfers, and to assess what scale of agricultural change would be needed to mitigate these transfers. We combine novel high-frequency phosphorus flux data from three representative catchments across the UK, a new high-spatial resolution climate model, uncertainty estimates from an ensemble of future climate simulations, two phosphorus transfer models of contrasting complexity and a simplified representation of the potential intensification of agriculture based on expert elicitation from land managers. We show that the effect of climate change on average winter phosphorus loads (predicted increase up to 30% by 2050s) will be limited only by large-scale agricultural changes (e.g., 20–80% reduction in phosphorus inputs).

[1] Lancaster Environment Centre, Lancaster University, Bailrigg, Lancaster, England LA1 4YQ, UK. [2] Rothamsted Research North Wyke, Okehampton, Devon, England EX20 2SB, UK. [3] Global Sustainability Institute, Anglia Ruskin University, Cambridge CB1 1PT England, UK. [4] Met Office Hadley Centre, Exeter, Devon, England EX1 3PB, UK. [5] University of East Anglia, Norwich, England NR4 7TJ, UK. [6] James Hutton Institute, Aberdeen, Scotland AB15 8QH, UK. [7] School of Engineering, Liverpool University, Liverpool, England L69 3GQ, UK. [8] Bangor University, Bangor, Gwynedd, Wales LL58 8RF, UK. [9] School of Computing, Mathematics & Digital Technology, Manchester Metropolitan University, Manchester M1 5GD, UK. [10] British Geological Survey, Keyworth, Nottingham, England NG12 5GG, UK. [11] School of Geographical Sciences, University of Bristol, Bristol BS8 1SS, UK. Correspondence and requests for materials should be addressed to M.C.O. (email: m.ockenden@lancaster.ac.uk) or to P.M.H. (email: p.haygarth@lancaster.ac.uk)

Climate change and the intensification of agricultural food production pose threats to water quality and aquatic ecosystem functions and services[1]. Biogeochemical flows, specifically phosphorus (P) and the one-way flow[2] of P from mineral reserves to farms and into oceans, are already considered to be beyond the safe operating space for sustainable human development[3]. Although intensification of food production may well be necessary from the standpoint of human demand and sustainable human development, this should also take account of societal concerns about resource use and eutrophication. Predicting future nutrient transfers into rivers, lakes and groundwater, for evaluating nutrient abatement strategies, is challenging, due to the complexity of the landscape processes involved and the uncertainties in the input data, model structures and calibration data. Previous studies on the effects of climate and land use change on water quality have been limited by inadequate data resolution, lack of appropriate P data, limited model comparison, and lack of uncertainty analysis[4–7]. Here, we overcome these limitations by combining new high-resolution catchment discharge and total P (TP) data and climate projections with two models of contrasting complexity (process-based Hydrological Predictions for the Environment (HYPE)[8] and a Data-Based Mechanistic (DBM)[9] model) for three diverse agricultural catchments across the UK. The Eden, Cumbria (predominantly livestock), the Wensum, Norfolk (predominantly arable) and the Avon, Hampshire (mixed farming) are representative of the country's different climatic conditions, soil types, hydrology and farming systems. We determine possible future agricultural management options by consultation with stakeholders in the three catchments. We estimate future P export loads under combined agricultural intensification and future climate using projections from a new high-resolution (1.5 km grid) regional climate model (RCM-1.5 km) for the UK[10] and from the UK Climate Projections 2009 Weather Generator[11] (UKCP09-WG). RCM-1.5 km is able to simulate sub-daily precipitation characteristics better than coarser resolution climate models, particularly for short-duration, summer convective storms[10], and makes a more realistic estimate of the intensity and duration of extreme events[12, 13] which have previously been shown to transfer a large proportion of the annual total P load[14, 15]. We assess what scale of agricultural change would be needed to mitigate the transfers predicted under climate change. Michalak[1] notes that climate research and water quality research are usually conducted entirely separately, partly due to the often differing scales of interest, and recommends that for better understanding of climate change effects, we need to bring together the two disciplines. Our integrated, multi-disciplinary study follows these recommendations, with the potential to contribute to the understanding of likely future P losses. We show that the predicted increase in winter P loads due to climate change (up to 30% by 2050s) is greater than the technically feasible reduction from mitigation measures estimated in previous studies[16]. Our study suggests that only large-scale agricultural changes (e.g., 20–80% reduction in P inputs) will limit the projected impacts of climate change on P loads in these catchments.

## Results

**Current phosphorus pollution sources and mitigation**. The current agricultural practices, management concerns, sources of pollution and current mitigation practices in each catchment, established from interaction with farmers, land managers and other stakeholders, are given in Table 1. For livestock-dominated catchments, the storage and spreading of organic livestock waste is a major concern, with inappropriate storage or insufficient storage capacity frequently resulting in farmers being forced to spread in suboptimal conditions, when the ground is frozen or saturated and the chance of heavy rainfall is high. The presence of heavy machinery on the land when the ground is wet can cause acute soil compaction, reducing infiltration and increasing the likelihood of surface runoff generation. For arable and horticulture-dominated catchments, diffuse pollution from nitrate and phosphate fertilisers is a major concern. In addition, soil erosion from roadside verges and field entrances, where frequent passage of farm machinery can damage the soil structure, results in sediment and nutrient laden road runoff when it rains. In both livestock and crop growing catchments, hard standings are identified as key sources of pollution, particularly where drain systems do not separate clean rainwater from dirty yard water.

Current strategies for mitigating phosphorus pollution depend on the key sources and on the hydrogeology of the catchment. In surface–water-dominated catchments, mitigation practices are currently aimed at breaking up the pollution transfer pathways. Hence, runoff detention features and settling ponds, designed to slow the flow and capture sediment and nutrients, are in current use. In groundwater-dominated catchments, mitigation practices

**Table 1 Major agricultural practices and pollution concerns for three catchments in the UK**

| Catchment | Dominant agricultural activities | Major agricultural concerns and key sources of pollution | Current mitigation practices |
|---|---|---|---|
| Newby Beck, Eden, Cumbria | Livestock grazing (cattle and sheep) Dairy production | Hard standings Slurry storage and management Inorganic fertiliser application Soil compaction | Runoff detention features |
| Blackwater, Wensum, Norfolk | Arable crops | Nitrate and phosphate fertilisers Runoff from road verges, hard standings, field entrances, eroding arable topsoils Soil denitrification (greenhouse gas emissions) Pesticide spraying Sewage Treatment Works | Cover crops Reduced cultivation measures Roadside sediment traps |
| Wylye, Avon, Hampshire | Livestock | Livestock waste management Inorganic fertiliser application Faecal pollution Soil erosion Septic tanks | Clean and dirty water separation Fencing watercourses Settling ponds |

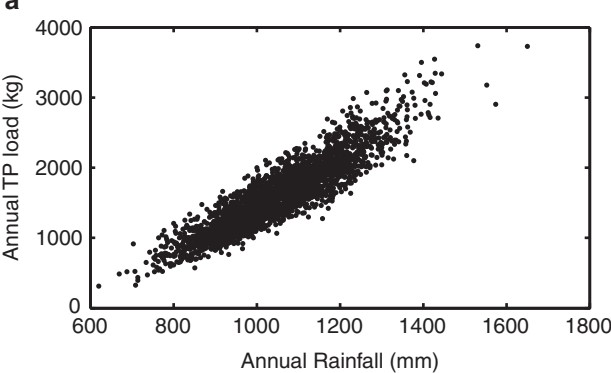

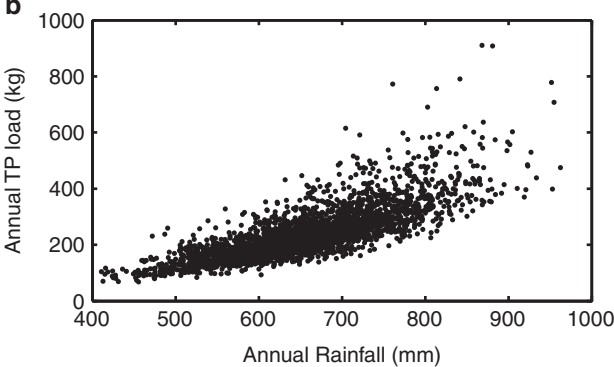

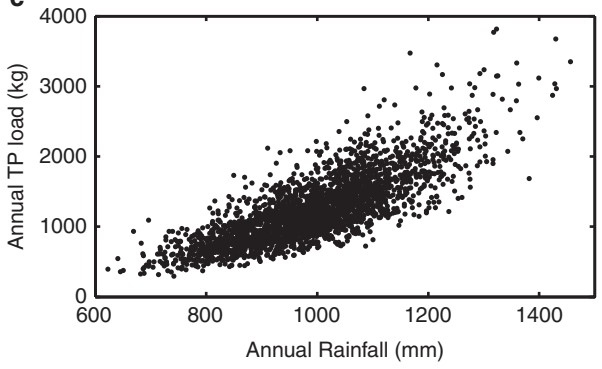

**Fig. 1** The relationship between annual total phosphorus load and rainfall in three UK catchments. Annual total phosphorus (TP) load and rainfall are from one behavioural parameter set in the Hydrological Predictions for the Environment (HYPE) model, for years 5–30 from 100 runs of baseline conditions for **a** Newby catchment, Eden ($R^2 = 0.82$, $p < 0.01$), **b** Blackwater catchment, Wensum ($R^2 = 0.61$, $p < 0.01$) and **c** Wylye catchment, Avon ($R^2 = 0.63$, $p < 0.01$). The dominant driver of annual P load is annual rainfall in all three of these diverse catchments

are aimed more at tackling sources and preventing mobilisation of sediment and phosphorus, using reduced cultivation measures and cover crops, or fencing streams to prevent livestock access.

**Streamflow and phosphorus loads under current conditions.** Using the parameters and models identified using the observed field data (Supplementary Data 3 (HYPE) and Supplementary Table 1 (DBM)), models are driven with the baseline climate data from UKCP09-WG to generate a range of P loads valid for current conditions (Supplementary Data 1). The dominant effect of rainfall in driving diffuse P loads from agricultural land is clearly shown by the relationship between annual rainfall and annual P load (Fig. 1), where annual rainfall explains between 61% (Wensum) and 82% (Eden) of the annual P load, including the

variability over 26 years of each model run. In spite of the non-linearity introduced by including a representation of the hydrological and soil processes in the process-based model HYPE (shown by deviation from the straight line in Fig. 1), this non-linearity is still relatively small compared to the dominant rainfall driver.

**Projections under climate change only.** Considering climate change alone initially, both the HYPE and DBM models predict very similar trends for future P exports, with increase in winter rainfall resulting in larger median winter flows and correspondingly larger winter P loads (up to 31% increase, Table 2). Larger changes in P load (25–31%) are expected in the wetter Eden and Avon catchments, than the Wensum (13–18%). In summer, decreases in median flow result in a decrease in median P load (6–21%). Seasonal changes in future rainfall are pronounced, with a 14–15% increase in median winter rainfall predicted by UKCP09-WG for the 2050s high emissions scenario across the Eden, Wensum and Avon catchments, and a 14–19% reduction in summer rainfall (Table 2). Larger percentage changes in rainfall, flow and P loads are shown with the high-resolution climate model, which may reflect both the better representation of extreme rainfall and a projection further into the future. However, as there is only one run of RCM–1.5 km, there can be no assessment of uncertainty[13]. These changes in rainfall patterns, including higher rainfall volumes and intensities, which have been projected by climate models for some time, are already being confirmed in observed rainfall records[17].

Total P loads from the HYPE and DBM models, driven by multiple runs of the UKCP09-WG, are shown in Fig. 2, with the high emissions scenario chosen as most representative of our current pathway. The inter-annual variability is very large, but, for both models, trends in the median winter P load show a clear increase for all catchments (by 2050s, around 30% in the Eden and Avon, 10% in the Wensum). In contrast, summer P exports show a decrease in all catchments (up to 20%), although the contribution of the summer load to the annual load is small, typically <15%, except in the drier Wensum catchment where there is less difference between summer and winter loads. For each of the catchments, the average observed seasonal P load, based on near-continuous sampling, is shown alongside the baseline model predictions. The single projection using the convection-permitting RCM-1.5 km is shown alongside the 2080s range, and generally lies within the uncertainty range estimated with the UKCP09-WG probabilistic projections. Compared to its baseline prediction, the P loads from RCM-1.5 km demonstrate the same trends as with UKCP09-WG, indicating that although the rainfall intensity is more realistically predicted, and particularly for summer convective storms[13], these storms do not make a significant difference to the median summer P exports. Although the results from RCM-1.5 km are not directly comparable to those using UKCP09-WG, because of the different time frame and the lack of uncertainty, they do not appear to show significant differences. However, the use of this extra climate model, giving results that are consistent with those from UKCP09-WG, adds further credence to the results.

As the estimations of change in phosphorus load from the DBM and HYPE models are similar, this suggests that the main mechanism driving the changes in phosphorus load is the change in seasonal rainfall totals. Other factors, such as temperature, which are included in HYPE but not in the DBM model, may also contribute, but this contribution is small compared to the dominant driver. Similarly, the lack of significant difference between results using the convection-permitting climate model and UKCP09-WG indicate that although rainfall intensity may

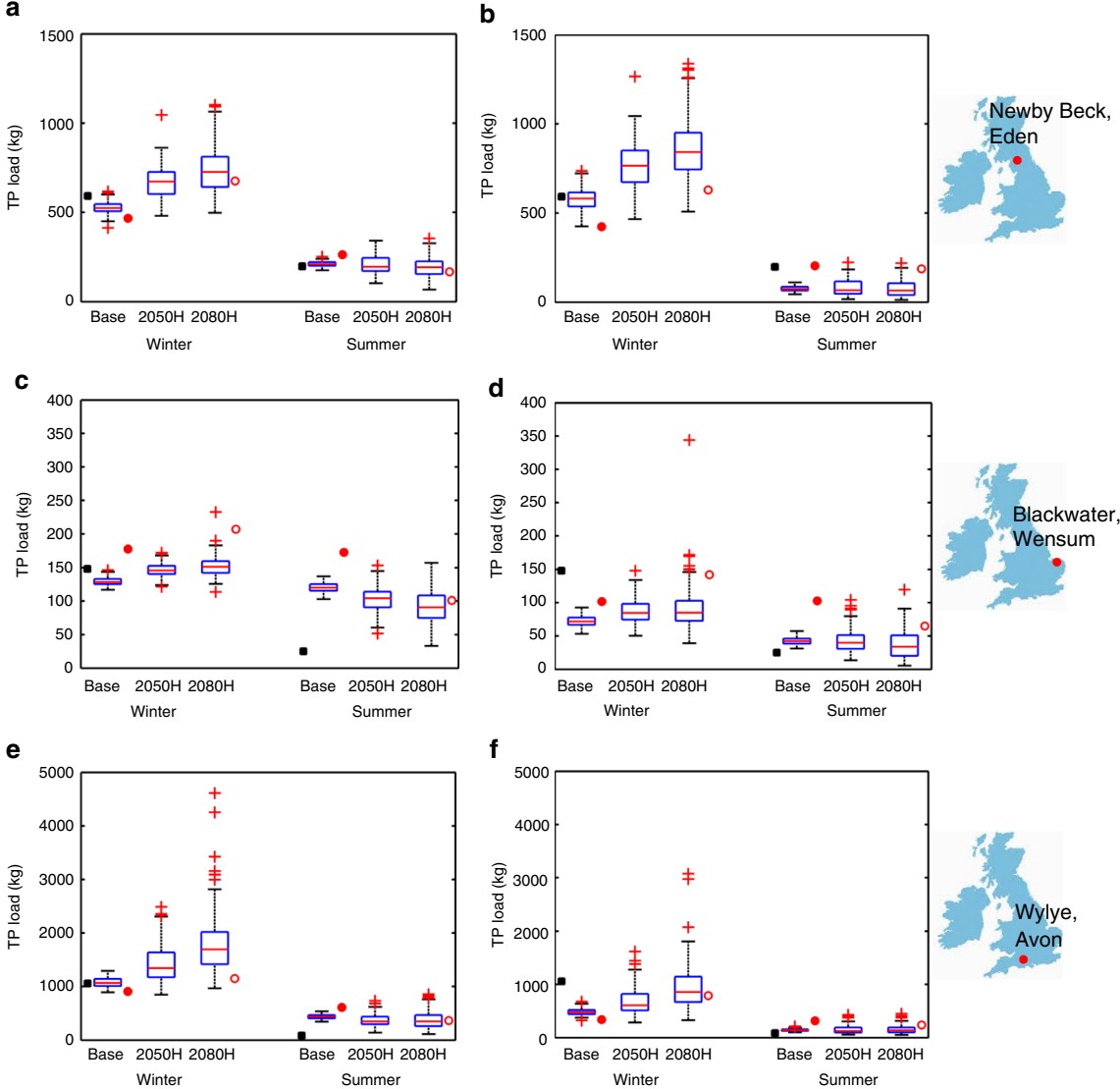

**Fig. 2** Range of likely seasonal total phosphorus loads exported from three UK catchments in the future. Box and whisker plots show the range of likely winter and summer total phosphorus (TP) loads predicted by Data-Based Mechanistic models (DBM) and process-based Hydrological Predictions for the Environment model (HYPE). TP loads for Newby Beck catchment, Eden, winter and summer from DBM **a** and HYPE **b**; TP loads for Blackwater catchment, Wensum, winter and summer from DBM **c** and HYPE **d**; TP loads for Wylye catchment, Avon, winter and summer from DBM **e** and HYPE **f** for present day conditions (Base), and future conditions representing 2050s (2040–2069) and 2080s (2070–2099) from UK Climate Projections 2009 Weather Generator, high emissions scenarios (2050H and 2080H respectively). HYPE results are mean of all runs with behavioural parameter sets. *Box* indicates inter-quartile range (25th–75th percentile), with median marked as *red line*; *whiskers* extend 1.5 times interquartile range beyond box, or to furthest data point if smaller. Outliers (beyond *whiskers*) are marked as *red +*. Observed winter and summer TP loads for 2012/13 are marked as *black squares*. Winter and summer TP loads predicted using rainfall from high-resolution climate model are marked as *red filled circles* (baseline conditions) and *red unfilled circles* (2100s)

also be a contributing factor, it is not as important as the change in rainfall volumes. All projections for flow and P load, for all emissions scenarios, including uncertainty and percentage changes are given in Supplementary Data 1.

**Projections under combined climate and agricultural change.** We combine the modelling presented above with a simple representation of future land management scenarios, which were determined following structured stakeholder elicitation workshops in each of the catchments. Stakeholders discussed and selected likely land management options from a range of possibilities. Most participants favoured the use of cover crops; this was strongest in the arable catchment, where this

management measure is already in use and has undergone both field testing and modelling of impact[18]. Soil conservation is of high importance in this catchment, where erosion of arable topsoil has been identified as a key concern (Table 1). In the more livestock-dominated catchments, an increase in winter housing for livestock and increase in slurry spreading were considered most likely. Both of these measures would affect the P loading on the soil, either in timing or quantity, and are likely to exacerbate the already identified concern of spreading livestock waste (Table 1). The percentage of stakeholders in agreement with these scenarios is given in Supplementary Fig. 1.

Spatial representations of specific land uses or land management changes are highly uncertain at a local scale[19], and semi-distributed, process-based models have been shown as not fit for

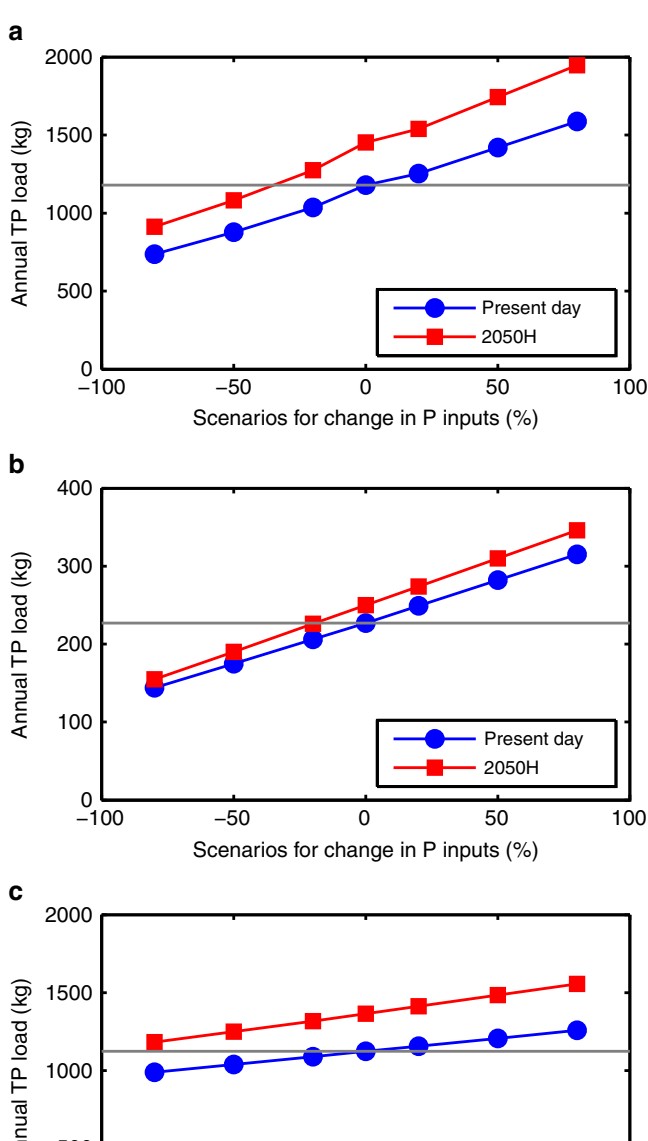

**Fig. 3** Variation in annual total phosphorus load under combined climate change and agricultural change scenarios for three UK catchments. Predicted changes to annual total phosphorus (TP) load from the mean of all behavioural runs from Hydrological Predictions for the Environment model (HYPE). Climate change scenario is 2050s high emissions (2050H) from UK Climate Projections 2009 Weather Generator. Agricultural change scenarios are represented by modifying P inputs by up to ±80%. Present day and 2050s high emissions scenario with up to ±80% change in P inputs for Newby Beck, Eden **a**, Blackwater, Wensum **b** and Wylye, Avon **c**. The *horizontal line* represents the present day annual load (mean of all behavioural runs) and the required level of reduction in the future. Large reductions in P (20–100%) are required to offset the projections of increased annual load due to climate change in the 2050s (high emissions)

purpose to model such detailed changes when data on parameters is lacking[20]. Therefore, we do not model any mitigation measures specifically; this is justified due to the inability of nutrient transfer models to include land management changes without large increases in uncertainty. Instead, we represent the agricultural

changes identified by expert elicitation as degrees of intensification of agricultural practices (+20, +50 and +80% increase in P inputs or equivalent reductions). An increase in P inputs represents, for example, increased application of fertilisers and manures, higher stocking densities or an increase in direct connectivity of sources to watercourses. Conversely, a decrease in P inputs can represent decreased application of in fertilisers or manures, more uptake and removal of phosphorus by plants or animals, or disruption of transfer pathways. This simple and transparent method of representing intensification enables the climate change impact to be evaluated both with and without management interventions, and without the confounding uncertainty associated with modelling land management changes.

Projections of P load using HYPE under climate change (2050s, high-emissions scenario) and changing intensification up to ±80% (Fig. 3) indicate that reductions in P inputs in all catchments would be required to offset the increase resulting from climate change. The model uses parameters based on the present day calibration, i.e., it assumes that initial soil P status in the future is the same as for present day. Using this assumption, a reduction in P inputs of at least 80% would be required in the Avon, around 20% reduction in the Wensum and 30% reduction in the Eden to maintain P losses from the catchments at the present day levels. The full range of modelled P input scenarios and all emissions scenarios are given in Supplementary Data 2. For the present day conditions, decreasing P inputs by 80% in the Avon makes only a 12% difference to the annual P load, compared to 37% in the Wensum and 38% in the Eden, meaning that a much larger reduction in P inputs is required in the Avon to offset the increase resulting from climate change. This reflects the different sources of P in the Avon, which are not all agricultural: there is a high background concentration of total P at low flows (around 0.12 mg l$^{-1}$). A significant proportion is due to sewage treatment works and to rural septic tanks. There may also be a minor contribution from apatite nodules in the Chalk which underlies this catchment[21].

Given the importance of rainfall for P transfers[14], and particularly in winter when catchments are generally more saturated, it may be prudent for policy advisors and land managers to prioritise mitigation measures that reduce stores of P (and other pollutants) in the soil and address winter runoff. This could also lessen the impact of extreme events on in-stream ecological communities, which can be reset by large geomorphological changes and nutrient transfers that may occur during flooding[22]. In summer, our model data suggest that the reduction in discharge will be accompanied by a comparable reduction in P loss, keeping concentration levels similar (Supplementary Fig. 2). However, where point sources contribute to P loads, unless these sources are also reduced, a reduction in discharge in summer could result in an increase in P concentration and greater risk of degradation in ecological status.

## Discussion

The effects of mitigation measures are hard to identify in observed data, because they are heavily masked by the inter-annual variability in rainfall, which translates to the large range in P loads under both simulated present day and future climate conditions (Fig. 2). The effects are further obscured in catchments where non-agricultural sources of P are a dominant contributor to poor water quality. However, overall trends in the flow regime between the present day and future are clear in Table 2 (with full uncertainty ranges in Supplementary Data 1), which shows large percentage increases in winter rainfall and discharge. Increases in rainfall volume and intensity both have the potential to increase phosphorus transfers, through increased

**Table 2 Future predicted percentage changes in hydrology and total phosphorus loads for three catchments in the UK**

| | Model | UKCP09-WG Winter %Δ | UKCP09-WG Summer %Δ | RCM-1.5 km Winter %Δ | RCM-1.5 km Summer %Δ |
|---|---|---|---|---|---|
| Newby Beck, Eden, Cumbria | Rainfall | +15 | −14 | +29 | −36 |
| | Discharge HYPE | +11 | −27 | +10 | −38 |
| | Discharge DBM | +27 | −7 | +40 | −34 |
| | Total P load HYPE | +31 | −8 | +49 | −9 |
| | Total P load DBM | +28 | −8 | +45 | −36 |
| Blackwater, Wensum, Norfolk | Rainfall | +15 | −14 | +14 | −47 |
| | Discharge HYPE | +3 | −16 | +7 | −38 |
| | Discharge DBM | +21 | −17 | +47 | −48 |
| | Total P load HYPE | +18 | −6 | +39 | −37 |
| | Total P load DBM | +13 | −13 | +16 | −42 |
| Wylye, Avon, Hampshire | Rainfall | +14 | −19 | +26 | −55 |
| | Discharge HYPE | +10 | −16 | +26 | −17 |
| | Discharge DBM | +25 | −20 | +25 | −39 |
| | Total P load HYPE | +27 | −15 | +134 | −24 |
| | Total P load DBM | +25 | −21 | +27 | −40 |

Seasonal percentage changes in median precipitation, discharge and total phosphorus (P) load predicted using two different models, Hydrological Predictions for the Environment model (HYPE) and Data-Based Mechanistic model (DBM). Climate data is for 2050s high emissions scenario from UK Climate Projections 2009 Weather Generator (UKCP09-WG), and for 2100 (from 1.5 km regional climate model, RCM-1.5 km), Winter=December, January, February; Summer=June, July, August

surface runoff and associated soil erosion. Also, increased recharge of deeper water reserves may result in more transfer of dissolved phosphorus forms. The dominance of high rainfall, as a driver of phosphorus load (Fig. 1) has been noted in other studies[14], particularly during high erodibility periods. The relationship could be used on its own as a simple estimator of future phosphorus load, (e.g., as in Ockenden et al.[15]), but the models used here improve on this simple estimation by including the non-stationarity of the relationship which results from the change in rainfall distribution in the future and the resulting change in effective rainfall.

In addition, measurable mitigation effects are buffered because stores of 'legacy' P (and also nitrogen) in the soil and other areas of catchments have accumulated over several decades[23]. These legacy stores present a challenge to water quality remediation[24] as it may take several more decades of little or no further P input before the stores are depleted to levels last seen before large-scale intensification of agriculture in the middle of the 20th Century. Reductions in P inputs (particularly if P becomes limiting) could have implications for agricultural production, although this may be compensated for by higher productivity of crops under future climate scenarios[25]. Indeed, the reduction in P inputs suggested here for sustainable water quality may not be compatible with the need for increased agricultural productivity and will require reassessment of priorities[26]. Due to the time lags in response, such as gradual depletion of soil P or the rate of adaptation by farmers[27], it is important to adopt an integrated approach to understanding climate effects on sustainable agriculture. Interactions between climate and agro-ecosystems are highly non-linear and changes to either will have feedback effects on the other[28]. This is particularly important for hydrology and nutrient transfer projections, where previous work has shown that the non-linearities might have different magnitudes of change depending on whether the climate projections were bias-corrected[29].

This study shows that the underlying trend of increasing P losses under climate change (up to 30% by 2050s) is larger than the theoretical reduction recently predicted for maximum uptake of farmer-preferred mitigation options (around 15% for catchment scale[16]). These findings are also applicable to other agricultural regions in the world with temperate climates where wetter winters are projected. Our analysis indicates that we would need to adopt large agricultural changes (e.g., 20–80% reduction

in P inputs) to counter the increased winter P losses projected by climate change. The example of P inputs has been used here to demonstrate the relative scale of climate change and land management change impacts. We have modelled changing P inputs, which can be easily interpreted, to represent many more spatially specific mitigation measures, which models are not capable of representing without big increases in uncertainty. We are not suggesting that P inputs should be the sole focus of mitigation measures, indeed we recognise that such measures need to be catchment specific, addressing sources, mobilisation and transfer along the transfer continuum[30]. There are many motives and challenges for farmers and stakeholders in choosing[31] or accepting[32] mitigation options. Stakeholders in this study are already quite well focussed on appropriate measures for their specific catchments, but we demonstrate that these measures may not be enough in the face of the climate challenge. At a catchment scale, currently adopted mitigation measures have not been able to realise a comparable magnitude of reduction to counter the increase projected by climate change.

## Methods

**Study catchments and field data**. The study catchments comprised three highly instrumented landscapes within the UK Defra Demonstration Test Catchments (DTC) Programme[33]: Newby Beck at Newby, Eden, Cumbria (54.59° N, 2.62° W; 12.5 km²); Blackwater at Park Farm, Wensum, Norfolk (52.78° N, 1.15° E; 19.7 km²); Wylye at Brixton Deverill, Avon, Hampshire (51.16° N, 2.19° W; 50.2 km²). Meteorological, flow and nutrient data are available at hourly resolution for parts of the period October 2011–September 2014. Further details of these catchments and the monitoring are available in Outram et al.[34].

**Hydrological modelling**. We applied two hydrological models of different complexity to the three DTC catchments. Parameters for both models were identified on all, or part, of the period 1 October 2011–30 September 2013, and validated on the period 1 October 2013–30 September 2014, using observed field data.

The Hydrological Predictions for the Environment (HYPE) model[8] is a hydrological model for simulating water flow and transport and turnover of nitrogen and phosphorus. The model is semi-distributed, dividing the landscape into classes according to soil type, land use and altitude. In agricultural lands the soil is divided into up to three layers, each with associated parameters. Soil, water and nutrient processes are simulated, with surface runoff, macropore flow, tile drainage and outflow calculated from individual soil layers. HYPE runs at a daily timestep, although the daily flows and phosphorus loads were calculated from the sub-daily observed data.

Nutrient inputs into the model in the form of fertiliser and manure were determined from the Defra British Survey of Fertiliser Practice. Nutrient inputs from point sources were added to each reach of the river or stream based on the

measured discharges and pollutant concentrations from the Environment Agency national register of consented discharges for the period 2010–2012, as used in a national source apportionment tool[35]. No changes were made to point sources for the future.

Parameters in the HYPE model were chosen using the Generalised Likelihood Uncertainty Estimation (GLUE) methodology[36] which samples the multi-dimensional parameter space to find sets of parameters which produce acceptable models which satisfy the evaluation criteria (termed behavioural models). While we recognise the sampling size is not extensive, given other experimental uncertainties we consider the chosen trajectories are indicative of the potential for change and in balance with all other computational and experimental design constraints. Models were evaluated using the Nash Sutcliffe Efficiency (NSE) coefficient[37] for discharge (Q) and for P. The thresholds for behavioural parameter sets were chosen to give the best 10–15 behavioural parameter sets. For Eden these were: NSE $\geq$ 0.6 for Q and NSE $\geq$ 0.5 for P; for Wensum these were NSE $\geq$ 0.55 for Q and NSE $\geq$ 0.53 for P; for Avon these were NSE $\geq$ 0.6 for Q and NSE $\geq$ 0.6 for P. The ranges for parameter sets used in the projections are given in Supplementary Data 3. The HYPE model calibration and validation fit statistics are given in Supplementary Data 4. HYPE allows simple user-specified changes to land use or management by variation of the input parameters.

Data-Based Mechanistic (DBM) modelling[38], using the CAPTAIN Toolbox for MATLAB[39] identified transfer function models for rainfall-runoff and rainfall-phosphorus load directly from the high temporal resolution (hourly) data, requiring very few parameters. Either discrete-time or continuous-time transfer function models[9], with the structures and parameters given in Supplementary Table 1, were identified directly from the hourly resolution observation data.

A second-order discrete-time linear transfer function with no noise model, denoted by [2, 2, $\delta$] takes the form:

$$y(t) = \frac{b_1 + b_2 z^{-1}}{1 + a_1 z^{-1} + a_2 z^{-2}} u(t - \delta) \tag{1}$$

where $y(t)$ is model output at time $t$, $u(t)$ is model input, $z^{-1}$ is the backwards step operator, i.e., $z^{-1}y(t) = y(t-1)$. $b_1$, $b_2$, $a_1$, $a_2$ are parameters determined during model identification and $\delta$ is the number of time steps of pure time delay. For a physical interpretation, models are only accepted if they can be decomposed by partial fraction expansion into two first-order transfer functions with structure [1, 1, $\delta$] representing fast and slow pathways, with characteristic time constants and steady state gains, i.e.

$$y(t) = \frac{b_f}{1 - a_f z^{-1}} u(t - \delta) + \frac{b_s}{1 - a_s z^{-1}} u(t - \delta) \tag{2}$$

where $b_f$ and $b_s$ are gains on the fast and slow pathways, respectively, and $a_f$ and $a_s$ are parameters characterising the time constants of the fast and slow pathways respectively. $a_f$ and $a_s$ are roots of the denominator polynomial in the second-order transfer functions above (Eq. (1)).

A second-order continuous-time linear transfer function with no noise model takes the form:

$$Y(s) = \frac{b_1 s + b_2}{s^2 + a_1 s + a_2} e^{-s\tau} U(s) \tag{3}$$

where, $Y(s)$ and $U(s)$ represent the Laplace transforms of the output and input, respectively. $b_1$, $b_2$, $a_1$, $a_2$ are parameters in the denominator and numerator polynomials in the derivative operator $s = \frac{d}{dt}$ that define the relationship between the input and the output, and $\tau$ represents the delay. Models are only accepted if they can be decomposed by partial fraction expansion into two parallel, first-order transfer functions, i.e.

$$Y = \frac{b_f}{s + a_f} e^{-s\tau} U + \frac{b_s}{s + a_s} e^{-s\tau} U \tag{4}$$

where $a_f$ and $a_s$ are direct reciprocals of the fast and slow time constants respectively, which define the fast and slow components of the response. $b_f$ and $b_s$ are parameters which determine the gain of the fast and slow components, respectively.

Note that parameters $b_1$, $b_2$, $a_1$, $a_2$ (and parameters $b_f$, $b_s$, $a_f$, $a_s$) have different interpretation, and therefore different values between discrete-time and continuous-time models. The relationship between the parameters (see most Control Engineering textbooks[40]) between discrete model denoted by superscript d and continuous time model denoted by superscript c is as follows:

for instance, for denominator parameter $a_f$

$$a_f^d = e^{-a_f^c \Delta t} \tag{5}$$

while for $b_f$ we have:

$$b_f^d = \frac{b_f^c}{a_f^c}\left(1 - e^{-a_f^c \Delta t}\right) \tag{6}$$

Models are evaluated according to the Nash Sutcliffe Efficiency (NSE)[37] and the Young Information Criterion (YIC)[38], an objective statistical measure which combines how well the model fits the data together with a measure of over-parameterisation.

Linear models, using observed rainfall, were identified first, and then improved where possible by use of the 'effective' rainfall. The rainfall-runoff non-linearity[41] was based on the storage state of the catchment, for which the discharge was used as a proxy, i.e.

$$Re(t) = R(t)(Q(t-1))^\beta \tag{7}$$

where $Re(t)$ is the effective rainfall at time $t$, $R$ is the observed rainfall, $Q$ is the observed discharge and $\beta$ is a constant exponent which is optimised on the observed data at the same time as model identification. For rainfall-phosphorus load models, linear models using observed rainfall were identified first, and then improved using the same effective rainfall relationship as identified with the rainfall-runoff model, with the effective rainfall generated one step ahead, using Eq. (7) and the simulated discharge at time $(t-1)$.

The DBM models are able to make best use of the high-frequency data to capture P dynamics which typically occur at time scales of hours rather than days. High temporal resolution measurements of nutrient dynamics have previously demonstrated that a daily time step is insufficient to capture sediment and P dynamics[42], resulting in an underestimation of export loads. Definitions, structure, and parameters for the models identified are provided in Supplementary Table 1, along with model fit statistics for the identification and validation periods (Supplementary Data 5).

Both the water quality model HYPE and the simple DBM model assume that relationships and processes identified on the basis of present day data will still hold in the future. However, the HYPE model, despite the uncertainty associated with the large parameter set, allows variation of the parameters to simulate changing environmental and management conditions, whereas the simple DBM model reduces parameter uncertainty but has no explicit way to change the identified parameters. We have used the DBM models to investigate the P response to climate change only.

**Future climate data**. We drive the models with future climatic rainfall and meteorological data from a convection-permitting (1.5 km grid) regional climate model (RCM-1.5 km), which is a configuration of the Met Office Unified Model[43], and with data from the UKCP09 Weather Generator[11], at hourly resolution for DBM and daily resolution for HYPE. The differences between UKCP09-WG and RCM-1.5 km are given in Supplementary Table 2

The UKCP09 Weather Generator[11] creates synthetic time series of weather variables at daily and hourly frequency, at 5 by 5 km grid square resolution, which are consistent with the underlying UKCP09 Climate Projections[44] (from an ensemble of 11 RCMs, 25 km grid). The Weather Generator applies stochastic models to generate many different, but statistically equivalent, time series which are stationary for a given time slice (30-year baseline or future time period) and emissions scenario.

The Weather Generator uses a stochastic rainfall model, with other variables generated according to the rainfall state, based on empirical relationships between climate variables in a baseline dataset of observations (1961–1995, 5 km grid). Future weather generator time series are generated by perturbing with change factors taken from the probabilistic projections of UKCP09. This ensures that the overall statistics (means and standard deviations) of the weather generator distributions are the same as those projected by UKCP09. The generated climate distributions include both the natural climate variability (including the significant spatial signature as observed in the 5 km grid baseline observations) and some uncertainty from the UKCP09 projections (as change factors are derived from the 11 member ensemble), giving a statistically-based distribution of climate at each chosen location, for each emissions scenario.

This study employed random number seeds to generate 100 plausible time series of 30 years (30 year baseline period and 30 year future time period), for each location, for each specified emissions scenario, as detailed below. The parameters used for generation of the UKCP09-WG time series data are given in Supplementary Table 3 (Newby Beck, Eden), Supplementary Table 4 (Blackwater, Wensum) and Supplementary Table 5 (Wylye, Avon).

RCM-1.5 km is much more computationally demanding and, therefore, we use results from two 12-year simulations: one for the period 1996–2009 and a 12-year simulation representative of the year 2100, under the RCP 8.5 scenario. The high resolution, convective permitting model allows more realistic representation of rainfall at a local scale, particularly for extreme events[13].

**Expert elicitation to guide future agricultural changes**. We determined likely future agricultural changes through expert elicitation with stakeholders at workshops held in each catchment. Discussion on identical topics in each catchment was followed by completion of a questionnaire (Supplementary Tables 6 and 7). The responses to agricultural change options (selected responses in Supplementary Fig. 1) guided the modelling of future changes. Much of the discussion of likely agricultural changes was indicative of intensification of agriculture, either through an increase in livestock numbers or the need to grow

more crops on the same land. The results of the discussions and the questionnaires were used to define simple land management change scenarios that could be easily incorporated into the HYPE model. Due to the semi-distributed nature of the model, it was not possible to make spatially explicit land use changes. Using lumped land use changes (not related to location, but related to the hydrological response unit) increased uncertainty in the result, as there was large variability in the result from different spatial representations. As the discussions and questionnaires pointed to intensification in farming, it was decided to include this in as simple and transparent a way as possible. The percentage changes for agricultural P inputs (±20, ±50, ±80% relative to baseline) are arbitrary, but cover a range of different scenarios for intensification. This simplified representation of likely agricultural changes could be applied transparently to all three catchments without the spatial uncertainty associated with more specific measures.

**Projections under climate change only**. We make future projections of discharge and P load under climate change only using both HYPE and DBM with the same parameter sets as identified with the observed field data. Using climate data from UKCP09-WG, annual loads (for baseline and scenario conditions) were calculated by taking the average over the last 26 years of each 30 year run, with the first 4 years being used as a spin-up period. For each 30 year run with HYPE, the averages were also taken over all behavioural runs. Winter loads were calculated by summing over the months December, January and February for each run. Summer loads were summed over June, July and August. 5th and 95th percentiles were calculated from the spread over 100 runs. Using climate data from RCM-1.5 km (12 year run), annual and seasonal loads were calculated by taking the mean over years 3–12.

**Projections under combined climate and agricultural change**. We made future projections of P load under combined climate change and agricultural change scenarios using HYPE. The agricultural scenarios, indicating a change in intensification of agriculture, were represented by increasing or decreasing the fertilizer and manure inputs (−80, −50, −20%, no change, +20, +50, +80%). No changes were made to point source inputs. Annual and seasonal loads were calculated as above.

**Data Availability**. The DTC data are available at http://www.environmentdata. org/dtc-archive-project/dtc-archive-project (Browse by Collection>Defra Collections>Eden/Hampshire Avon/Wensum Demonstration Test Catchment Data). UKCP09 data and access to the Weather Generator is available at http://ukclimateprojections.metoffice.gov.uk/.

The data underlying the figures in this manuscript are openly available from Lancaster University data archive at https://dx.doi.org/10.17635/lancaster/researchdata/110.

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

## Acknowledgements

This work was funded by the Natural Environment Research Council (NERC) as part of the Nutrients in Catchments to 2050 project (NUTCAT 2050), grants NE/K002392/1, NE/K002430/1 and NE/K002406/1, and supported by the Joint UK BEIS/Defra Met Office Hadley Centre Climate Programme (GA01101). We are grateful to the UK Demonstration Test Catchment (DTC) research platform (Defra projects WQ0210, WQ0211, WQ0212 and LM0304) for provision of the field data.
The UK Climate Projections (UKCP09, © Crown Copyright 2009) have been made available by the Department for Environment, Food and Rural Affairs (Defra) and the Department of Energy and Climate Change (DECC) under licence from the Met Office, UKCIP, British Atmospheric Data Centre, Newcastle University, University of East Anglia, Environment Agency, Tyndall Centre and Proudman Oceanographic Laboratory. These organisations give no warranties, express or implied, as to the accuracy of the UKCP09 and do not accept any liability for loss or damage, which may arise from reliance upon the UKCP09 and any use of the UKCP09 is undertaken entirely at the users risk.

## Author contributions

M.C.O. ran the DBM model and led the writing of the paper. M.J.H. ran the HYPE model. M.L.V. set up HYPE model parameter sets. W.T. assisted with DBM modelling. C.W. collated the stakeholder information. R.K. ran the 1.5 km climate model for the Northern UK. M.C.O., P.M.H., M.J.H., C.W., K.J.F., P.D.F., R.K., A.L.C., C.J.A.M., K.M.H., R.E. facilitated at stakeholder workshops. P.M.H. was overall project lead with K.J.B., P.D.F., P.J.A.W. and J.G.Z. also helping manage the project. All authors participated in interpretation of results and the writing and editing process.

## Additional information

**Competing interests:** The authors declare no competing financial interests.

