## [Peer Review File · Nature Communications]

Reviewers' Comments:

Reviewer #1 (Remarks to the Author)

This is a very interesting study that employs a combination of techniques to draw out meaningful conclusions about climate change and watershed phosphorus mitigation. While I am aware of other studies using watershed models to forecast changes in nutrient loads with climate change, I am not aware of any that employ the high temporal resolution models and, more importantly, the stakeholder derived mitigation strategies, that are used in this study. This is a very innovative study with high potential for impact.

With that said, I do think the authors could substantially improve their manuscript to (a) make it more meaningful to readers interested in what agriculture can do to address societal concerns of eutrophication, (b) address the question of whether the stakeholder committees were actually on the right track when they recommended future management strategies. This can be done by focusing the initial modeling (using historical data) on defining the nature of the problem in these watersheds. What are the major management concerns, what are the likely sources of pollution, are there critical source areas? This will require minimizing the discussion on runoff driven trends, but there is room to do that. It will also help to answer the question as to why mitigation strategies won't work (because the stakeholders were not sufficiently ambitious in proposing management strategies or because not management option exists)?

Again, this is an excellent piece that, I think, paves the way for interactive, multidisciplinary watershed studies. But, it could be improved.

Specific comments

P3 - note that sustainable intensification is the theme of research and food production strategies aimed at meeting a growing global population. Therefore, intensification may very well be necessary from the stand point of human demand and the "safe operating space for sustainable human development" (not beyond it)

P41st paragraph - suggest your replace "agricultural change options" with "agricultural management options"

P4-P5 paragraphs (considering climat change alone...). These paragraphs should be modified to include some statements about the major current opportunities for mitigating water quality. You should be able to discuss what types of soils management variables were most important to today's P export. There is no sense here as to what type of agriculture you are evaluating and what is currently an option (or concern) for farmers.

P6 - "most participants favored the use of cover crops". This would imply that erosion is the dominant concern for P since this practice is geared toward soil conservation (as well as nitrogen capture). There is no indication of whether that is the primary concern in these watershed.

P6 - "in the more livestock-dominated catchments..." This would imply that practices aimed at dissolved P loss or long-term soil P loading are most important. Are the stakeholders on target?

Reviewer #2 (Remarks to the Author)

Authors have prepared a paper on mitigating phosphorus losses under climate change, focusing on changes in agriculture. Two models were developed for three experimental watersheds in the UK

and then used to simulate impact of climate change (2050) under different mitigation measures.

The paper is well written but somehow hard to follow due to missing headings.

There isn't enough information provided about the model fit to understand uncertainty of the model outputs. Currently, only NSE criteria are shown in S3 but not the actual performance values. I would recommend the authors also add relative error or PBIAS to supplemental information, both for the full calibration & validation periods as well as for the evaluated seasons (summer, winter). Statistics should be given for flow, TP concentration, and TP load for each watershed. Median and ranges from the individual realisations should be provided. The observed winter and summer loads are plotted in Figure 1 but without any quantitative evaluations of the fit.

I. 113: it is unclear what the authors' intention here is. Please describe in more details what you mean by "P inputs." Does this mean that the proposed measures identified by the stakeholders were only simulated as changes in fertilizer amounts as stated in methods? If this is the case, it represents a rather significant limitation in the usability of the study and a disconnect from the stakeholders' questionnaire.

I. 155 and others (discussion of the impact). It would be helpful if the authors describe how changes in the flow regime itself affected the P transport. P loads are highly driven by high precipitation events occurring during high erodibility periods.

I. 176-178: This seems contrary to the previous statements that even high reductions in P inputs are not able to mitigate the increase due to climate change. Perhaps the focus of the mitigation measures should be not only on P inputs but also other aspect of hydrological cycle, erosion, and sediment transport.

Reviewer #3 (Remarks to the Author)

"Major agricultural changes required to mitigate phosphorus losses under climate change" by Ockenden et al., submitted to Nature Communications

Review by Hoshin V Gupta, Professor, University of Arizona

Summary of the Main Message: This communication reports on the results of a modeling study, based on stakeholder inputs, to evaluate the impact of projected (2050 and beyond) climate change on land to water Phosphorus losses at three agricultural catchments in the UK, and to assess what scale of agricultural change would be needed to mitigate these transfers. The results suggest that average winter P loads can be expected to increase up to 30% by the 2050s and that large-scale agricultural changes resulting in 20-80% reduction in P inputs will be needed to limit the projected impacts, which is greater than considered technically feasible by use of farmer-preferred mitigation measures. The study recommends a) the prioritization of mitigation measures that reduce stores of P (and other pollutants) in the soil, and b) reduction in point sources that contribute to P loads. Because 'legacy' stores of 'P (and nitrogen) have accumulated over decades, it may take several decades of reduced P input before the stores are depleted to mid-20th Century levels. This has implications for agricultural production, and may not be compatible with the need for increased agricultural productivity, requiring reassessment of priorities.

Reviewer Comments:

1) Overall: Overall, the study seems well conceived and executed, and the main message is clear (although could be enhanced as indicated below).

2) Presentation: I found the report a little difficult to read due to the manner in which the material was presented (i.e., organization), which also suffers from some degree of repetition.

To gain a clear understanding, I found it useful to rearrange the presentation in the manner provided below (with some details omitted). I include this for the authors benefit, along with my summary above, in case they find it helpful in revising the manuscript to improve the presentation of their message and thereby increase its impact.

3) **Hydrological Modeling:** I was asked to specifically assess the hydrological modeling aspect of the study. Given the brief nature of this communication, very little actual modeling detail is provided and therefore I am not able to say very much substantive in this regard. Overall, the fact that both a semi-distributed process-based daily time step model and a hourly time step databased mechanistic model give similar results suggests that the modeling results are probably reasonable, but this may also be a result of the fact that there is a clear and extremely strong relationship between rainfall and P load (Figure 3), at the annual level, and so the results likely do not depend strongly on the choice of hydrological model (except in degree). It might be interesting, therefore, for the authors to compare their results to a back-of-the-envelope calculation based on only the explicit relationship shown in Figure 3, and to comment on any differences that might show up between the detailed modeling approach and a crude 'quick and dirty' assessment.

4) **Climate Modeling:** Similarly, it might be useful to clearly remark on whether there was any significant difference in assessed results using the new high-resolution convection-permitting regional climate model over the other approaches tested (I gather not, based on comments in the text). That is not to say that the convection-permitting model results are not useful, since they add confidence/credibility to the overall results.

5) Such remarks (re hydrological and climate modeling) are useful in generally evaluating the broad policy-relevant benefits of very detailed modeling approaches (as opposed to evaluating specific local impacts and remediation strategies).

6) Towards the end the communication alludes to the "recommendations of Michalak" without clarifying for the general reader what those recommendations actually were/are. It would therefore be useful to do so.

7) It would be useful to clarify that the main climate change mechanism resulting in the P load changes is specifically changes in seasonal rainfall and not (if I understood correctly) other factors such as temperature etc. Further, please clarify whether it is simply increased amounts of rainfall or changes in rainfall intensities that is the controlling factor, or both. The term "climate change" is so general as to be almost non-informative when trying to understand how to respond in setting policy and mitigation strategies, so a little more specificity could be very helpful.

SUMMARY OF MAIN POINTS IN THE PRESENTATION

The Problem

- 1) Climate change and intensification of agricultural food production pose threats to water quality and aquatic ecosystem functions and services.
- 2) Biogeochemical flows (specifically P) are already considered to be beyond the safe operating space for sustainable human development, and nutrient abatement strategies are needed.
- 3) Predicting future nutrient transfers into rivers, lakes and groundwater is challenging, due to the complexity of the landscape processes involved and the uncertainties in the input data, model structures and calibration data.
- 4) Previous studies have been limited by inadequate data resolution, lack of appropriate P data, limited model comparison, and lack of uncertainty analysis.

Study Goals

- 1) To use a combination of modeling and stakeholder-based inputs to evaluate the impact of projected climate change (to 2050 and beyond) on land to water Phosphorus (P) losses at three agricultural catchments in the UK
- 2) To assess what scale of agricultural change would be needed to mitigate these transfers.
- 3) To overcome the aforementioned limitations of previous studies.

Approach

To combine:

- 1) P flux data from three representative catchments across the UK
- 2) Projections provided by a new high spatial resolution climate model
- 3) Uncertainty estimates provided by an ensemble of future climate simulations
- 4) P transfer models of contrasting complexity
- 5) A simplified representation of the potential intensification of agriculture based on expert elicitation from land managers.

Methods

- 1) Models were developed using meteorological, flow and nutrient data available at hourly resolution for Oct 2011 – Sept 2014 (three years).
- 2) Two hydrological models of different complexity were applied; a) the process-based, semi-distributed water quality model HYPE run at a daily time step, and b) a databased continuous-time transfer function mechanistic model run at hourly time step. Model parameters were identified using the first two years and evaluated using the third year.
- 3) Both models were used to make future projections of discharge and P load under climate change. Future climatic rainfall and meteorological data were taken from three sources; a) a convection-permitting regional climate model (RCM-1.5km), b) the Met Office Unified Model, and c) data from the UKCP09 Weather Generator. For the UKCP09-WG, a 30 year probabilistic ensemble (100 runs) was generated for both baseline and scenario conditions.
- 4) Average annual loads were calculated using 26 years, with first four years as spin-up. For HYPE, results were averaged over all behavioral parameter sets. Winter loads were computed for Dec/Jan/Feb, and summer loads for Jun/Jul/Aug. Uncertainty estimates are based on the spread over 100 runs. For RCM-1.5km only 12-year simulations were used.
- 5) Future agricultural changes were determined via expert elicitation with stakeholders in each catchment. And used to guide modeling of future changes.
- 6) Future projections of P load were made under combined climate change and agricultural change scenarios using HYPE, by increasing or decreasing fertilizer and manure inputs, with no changes to point source inputs.

Results

Current conditions:

- 1) There is a clear relationship between annual rainfall and annual P load (Figure 3), illustrating the dominant effect of rainfall in driving diffuse P loads from agricultural lands

Considering climate change alone (2050s, high emissions scenario):

- 2) Pronounced seasonal changes in future rainfalls are predicted for the 2050s high emissions scenario (Table 1).
 - a. 14-15% increase in median winter rainfall
 - b. 14-19% reduction in summer rainfall
 - c. The higher resolution climate model shows even larger percentage changes
- 3) Inter-annual variability is very large, but both models predict very similar trends for future P exports (Figure 1).
 - a. Increased winter rainfall resulting in larger median winter flows and correspondingly larger (up to 31% increase) winter P loads.
 - b. While summer decreases in median flow result in a (6-21%) decrease in median P load, the summer contribution to the annual load is small (typically less than 15%)

Considering climate change combined with future land management scenarios:

- 4) In the arable catchment, stakeholders favored the use of cover crops, while in the livestock-dominated catchments they favored increase in winter housing for livestock and increase in slurry spreading.
- 5) Projections of P load (Figure 2) indicate that reductions in P inputs would be required to offset the increase resulting from climate change.
- 6) Much larger P input reductions are required in the Avon, reflecting the different sources of P, which are not all agricultural (high background concentrations, sewage treatment works, rural

septic tanks, apatite nodules in the underlying Chalk).

Conclusions

For the studied catchments:

- 1) The integrated approach used here supports the recommendations of Michalak and contributes to the understanding of likely future P losses.
- 2) Land to water P losses will be impacted by climate change and land management for food production, with detrimental impacts on aquatic ecosystems.
- 3) Climate change is predicted to increase average winter P loads up to 30% by 2050s,
- 4) Large-scale agricultural changes that result in 20-80% reduction in P inputs will be needed to limit the projected impacts of climate change on P loads.
- 5) This is greater than the technically feasible reduction (around 15% for catchment scale) from farmer-preferred mitigation measures estimated by previous studies.
- 6) Given the importance of rainfall for P transfers (particularly in winter when catchments are generally more saturated), it may be prudent for policy advisors and land managers to prioritize mitigation measures that reduce stores of P (and other pollutants) in the soil and address winter runoff. This could also lessen the impact of extreme events on in-stream ecological communities, which can be reset by large geomorphological changes and nutrient transfers that may occur during flooding.
- 7) In summer, unless point sources that contribute to P loads are reduced, reduced discharge could result in increased P concentrations and greater risk of ecological degradation.
- 8) 'Legacy' stores of 'P (and nitrogen) accumulated over decades present a challenge to water quality remediation. It may take several decades of reduced P input before the stores are depleted to levels last seen in the mid-20th Century. This could have implications for agricultural production, and may not be compatible with the need for increased agricultural productivity, requiring reassessment of priorities.
- 9) Due to the response lag times, and non-linear interactions between climate and agro-ecosystems, it is important to adopt an integrated approach to understanding climate effects on sustainable agriculture.
- 10) These findings are applicable to other agricultural regions in the world with temperate climates where wetter winters are projected.

Note: page and line numbers refer to the revised, unmarked manuscript. Changes are shown with explanatory comments in the marked manuscript

Reviewers' comments:

Reviewer #1 (Remarks to the Author):

This is a very interesting study that employs a combination of techniques to draw out meaningful conclusions about climate change and watershed phosphorus mitigation. While I am aware of other studies using watershed models to forecast changes in nutrient loads with climate change, I am not aware of any that employ the high temporal resolution models and, more importantly, the stakeholder derived mitigation strategies, that are used in this study. This is a very innovative study with high potential for impact.

With that said, I do think the authors could substantially improve their manuscript to (a) make it more meaningful to readers interested in what agriculture can do to address societal concerns of eutrophication, (b) address the question of whether the stakeholder committees were actually on the right track when they recommended future management strategies.

This can be done by focusing the initial modeling (using historical data) on defining the nature of the problem in these watersheds. What are the major management concerns, what are the likely sources of pollution, are there critical source areas? This will require minimizing the discussion on runoff driven trends, but there is room to do that. It will also help to answer the question as to why mitigation strategies won't work (because the stakeholders were not sufficiently ambitious in proposing management strategies or because not management option exists)?

Again, this is an excellent piece that, I think, paves the way for interactive, multidisciplinary watershed studies. But, it could be improved.

Thank you. To address comment a) on what agriculture can do to address societal concerns of eutrophication, we have used the stakeholder discussions and knowledge of the catchments to add a new section defining the key management concerns and likely sources of pollution in each catchment. We have added a table detailing these (see specific changes detailed below), in order to improve readers' understanding of the pressures within the different catchments, and the pollution mitigation measures currently in use.

In addition, we have included an accompanying manuscript (about to be submitted) "Data-based mechanistic model of catchment phosphorus load improves predictions of storm transfers and annual loads in surface waters". This gives further background material on the data-based mechanistic (DBM) model development and calibration. The DBM model allows interpretation of the (quite different) dominant modes for each catchment, and thus identifies which phosphorus transfer pathways are likely to be

most important for mitigation measures. This helps to define the nature of pollution problems in each catchment, and thus which mitigation measures are likely to be most effective.

To address comment b) we have added comments on the use of certain measures for identified sources - p5 L96 (or comment MCO5 in marked text):

“Current strategies for mitigating phosphorus pollution depend on the key sources and on the hydrogeology of the catchment. In surface-water dominated catchments, mitigation practices are currently aimed at breaking up the pollution transfer pathways. Hence, runoff detention features and settling ponds, designed to slow the flow and capture sediment and nutrients, are in current use. In groundwater dominated catchments, mitigation practices are aimed more at tackling sources and preventing mobilization of sediment and phosphorus, using reduced cultivation measures and cover crops, or fencing streams to prevent livestock access.”

We have also added a comment in the discussion p12 L270 (or comment MCO16 in marked text):

“There are many motives and challenges for farmers and stakeholders in choosing or accepting mitigation options. Stakeholders in this study are already quite well focussed on appropriate measures for their specific catchments, but we demonstrate that these measures may not be enough in the face of the climate challenge.”

Whilst we do not model any mitigation measures specifically, we justify our decision not to do this on the basis of the inability of nutrient transfer models to include land management changes without large increases in the number of parameters and resulting uncertainty (see e.g. Dean et al., 2009, Stoch. Environ. Res. Risk Assess). Our simple and transparent method of representing intensification enables the climate change impact to be seen both with and without management interventions. The specific changes are detailed below.

Specific comments

P3 - note that sustainable intensification is the theme of research and food production strategies aimed at meeting a growing global population. Therefore, intensification may very well be necessary from the stand point of human demand and the "safe operating space for sustainable human development" (not beyond it)

We accept this point and have added a sentence on p3 L44 to acknowledge it:

“Although intensification of food production may well be necessary from the standpoint of human demand and sustainable human development, this should also take account of societal concerns about resource use and eutrophication.”

P4 1st paragraph - suggest your replace "agricultural change options" with "agricultural management options"

Accepted. P4 L59 changed to "agricultural management options"

P4-P5 paragraphs (considering climate change alone...). These paragraphs should be modified to include some statements about the major current opportunities for mitigating water quality. You should be able to discuss what types of soils management variables were most important to today's P export. There is no sense here as to what type of agriculture you are evaluating and what is currently an option (or concern) for farmers.

We have included a new sub-section p5 L80 with heading "Current phosphorus pollution sources and mitigation" including a table detailing the types of agriculture, key concerns and current mitigation practices in each catchment. We have added more explanation of the current mitigation practices and opportunities:

"Current phosphorus pollution sources and mitigation

The current agricultural practices, management concerns, sources of pollution and current mitigation practices in each catchment, established from interaction with farmers, land managers and other stakeholders, are given in Table 1. For livestock dominated catchments, the storage and spreading of organic livestock waste is a major concern, with inappropriate storage or insufficient storage capacity frequently resulting in farmers being forced to spread in suboptimal conditions, when the ground is frozen or saturated and the chance of heavy rainfall is high. The presence of heavy machinery on the land when the ground is wet can cause acute soil compaction, reducing infiltration and increasing the likelihood of surface runoff generation. For horticulture dominated catchments, diffuse pollution from nitrate and phosphate fertilisers is a major concern. In addition, soil erosion from roadside verges and field entrances, where frequent passage of farm machinery can damage the soil structure, results in sediment and nutrient laden road runoff when it rains. In both livestock and crop growing catchments, hard standings are identified as key sources of pollution, particularly where drain systems do not separate clean rainwater from dirty yard water.

Current strategies for mitigating phosphorus pollution depend on the key sources and on the hydrogeology of the catchment. In surface-water dominated catchments, mitigation practices are currently aimed at breaking up the pollution transfer pathways. Hence, runoff detention features and settling ponds, designed to slow the flow and capture sediment and nutrients, are in current use. In groundwater dominated catchments, mitigation practices are aimed more at tackling sources and preventing mobilization of sediment and phosphorus, using reduced cultivation measures and cover crops, or fencing streams to prevent livestock access."

Table 1 Major agricultural practices and pollution concerns for three catchments in the UK

Catchment	Dominant agricultural	Major agricultural concerns and key sources of pollution	Current mitigation practices
------------------	------------------------------	---	-------------------------------------

	activities		
Newby Beck, Eden, Cumbria	Livestock grazing (cattle and sheep) Dairy production	Hard standings Slurry storage and management Inorganic fertiliser application Soil compaction	Runoff detention features
Blackwater, Wensum, Norfolk	Arable crops	Nitrate and phosphate fertilisers Runoff from road verges, hard standings, field entrances, eroding arable topsoils Soil denitrification Pesticide spraying Sewage Treatment Works	Cover crops Reduced cultivation measures Roadside sediment traps
Wylde, Avon, Hampshire	Livestock	Livestock waste management Inorganic fertiliser application Faecal pollution Soil erosion Septic tanks	Clean and dirty water separation Fencing watercourses Settling ponds

In the discussion section we have added comments on the motives and challenges for stakeholders in accepting mitigation measures:

P12 L268 “We are not suggesting that P inputs should be the sole focus of mitigation measures, indeed we recognise that such measures need to be catchment specific, addressing sources, mobilisation and transfer along the transfer continuum²⁹. There are many motives and challenges for farmers and stakeholders in choosing³⁰ or accepting³¹ mitigation options. Stakeholders in this study are already quite well focussed on appropriate measures for their specific catchments, but we demonstrate that these measures may not be enough in the face of the climate challenge.”

P6 - "most participants favored the use of cover crops". This would imply that erosion is the dominant concern for P since this practice is geared toward soil conservation (as well as nitrogen capture). There is no indication of whether that is the primary concern in these watershed.

In addition to Table 1 and associated comments (see above) we have added a sentence after "most participants favored the use of cover crops..." (was p6, now p8 L169) “Soil conservation is of high importance in this catchment, where erosion of arable topsoil has been identified as a key concern (Table 1).”

P6 - "in the more livestock-dominated catchments..." This would imply that practices aimed at dissolved P loss or long-term soil P loading are most important. Are the stakeholders on target?

In addition to Table 1 and associated comments (see above) we have added a sentence (was p6, now p8 L173) “Both of these measures would affect the P loading on the soil,

either in timing or quantity, and are likely to exacerbate the already identified concern of spreading livestock waste (Table 1).”

Reviewer #2 (Remarks to the Author):

Authors have prepared a paper on mitigating phosphorus losses under climate change, focusing on changes in agriculture. Two models were developed for three experimental watersheds in the UK and then used to simulate impact of climate change (2050) under different mitigation measures.

The paper is well written but somehow hard to follow due to missing headings.

Subheadings have been added to Results and Methods sections

There isn't enough information provided about the model fit to understand uncertainty of the model outputs. Currently, only NSE criteria are shown in S3 but not the actual performance values. I would recommend the authors also add relative error or PBIAS to supplemental information, both for the full calibration & validation periods as well as for the evaluated seasons (summer, winter). Statistics should be given for flow, TP concentration, and TP load for each watershed. Median and ranges from the individual realisations should be provided. The observed winter and summer loads are plotted in Figure 1 but without any quantitative evaluations of the fit.

We have added two tables to the Supplementary Information with model fit statistics for HYPE (SI Table S4) and DBM (SI Table S6). These tables of model fit statistics include the observed and modelled median (Q50 or TP50), and ranges (Q10, Q90 or TP10, TP90) for discharge Q and total phosphorus load TP. Nash Sutcliffe Efficiency (NSE) and model bias (PBIAS) are given for the calibration period and the validation period.

Because the models were calibrated on the whole of the calibration period (and not specifically conditioned for winter or summer periods), we believe that the seasonal statistics or statistics for variables that were not calibrated (TP concentration) are misleading and are not meaningful in the context of the message of this paper. We believe that these seasonal statistics, for which the model was not specifically calibrated, also reflect the shortcomings of NSE and PBIAS as model fit statistics – because NSE is based on the square of model residuals, it tends to be dominated by how well the model fits the peaks, which, for both discharge and total phosphorus load, tend to be higher and more frequent in winter. This frequently leads to overestimation in summer, where a (relatively) small absolute error on a very small quantity results in a very large percentage error (and hence misleading PBIAS). However, we have provided the seasonal statistics for the reviewers in Tables S4 and S6 (expanded versions of the tables provided in the Supplementary Information) at the end of this response. For the reasons outlined above, we have also provided Figures S3-S14 below, to be viewed in

conjunction with the table S4 below. The figures show the observed and HYPE modelled data for specified 'best' and 'worst' behavioural runs, in order to illustrate the fit statistics in the context of the whole of the fit period.

We also reiterate that, since all results in this paper are based on the changes between modelled baseline data and modelled scenario data, any bias in the model is included in both the baseline results and the scenario results. Therefore, it will be much less important when considering changes only. However, we also acknowledge in the manuscript that non-linearities in nutrient transfer may have different magnitudes of change depending on whether climate bias is removed or not (p11 L254).

The observed winter and summer loads in former Figure 1 (now Figure 2) are not plotted against the calibrated loads, but against the modelled loads using the baseline data, which represents the variability in current climate. This is a qualitative illustration of how well the observed data fits within the ranges predicted, but the quantitative measures (which we believe are not meaningful without the explanation and the context provided by tables and figures below) are provided for the reviewers in the tables and figures at the end of this response.

1. 113: it is unclear what the authors' intention here is. Please describe in more details what you mean by "P inputs." Does this mean that the proposed measures identified by the stakeholders were only simulated as changes in fertilizer amounts as stated in methods? If this is the case, it represents a rather significant limitation in the usability of the study and a disconnect from the stakeholders' questionnaire.

We have clarified that we have used changes in P inputs to represent a range of changes in land management, but we disagree that this is a limitation in the usability of the study. We justify that the transparent way we have included changes in land management is not clouded by the additional uncertainty that ensues from trying to represent spatial land management changes in models which are incapable of doing this, or where doing so would increase parameter uncertainty even more than it is without the land management changes. We have added (p9 L180):

"Therefore, we do not model any mitigation measures specifically; this is justified due to the inability of nutrient transfer models to include land management changes without large increases in uncertainty. Instead, we represent the agricultural changes identified by expert elicitation as degrees of intensification of agricultural practices (+20, +50 and +80% increase in P inputs or equivalent reductions). An increase in P inputs represents, for example, an increase in fertilisers and manures, higher stocking densities or an increase in direct connectivity of sources to water courses. Conversely, a decrease in P inputs can represent a decrease in fertilisers or manures, more uptake and removal of phosphorus by plants or animals, or disruption of transfer pathways. This simple and transparent method of representing intensification enables the climate change impact to be evaluated both with and without management interventions, and without the confounding uncertainty associated with modelling land management changes."

l. 155 and others (discussion of the impact). It would be helpful if the authors describe how changes in the flow regime itself affected the P transport. P loads are highly driven by high precipitation events occurring during high erodibility periods.

We have added (p11 L228):

“However, overall trends in the flow regime between the present day and future are clear in Table 2 (with full uncertainty ranges in SI Table S1), which shows large percentage increases in winter rainfall and discharge. Increases in rainfall volume and intensity both have the potential to increase phosphorus transfers, through increased surface runoff and associated soil erosion. Also, increased recharge of deeper water reserves may result in more transfer of dissolved phosphorus forms. The dominance of high rainfall as a driver of phosphorus load (Figure 1) has been noted in other studies¹⁴, particularly during high erodibility periods. The relationship could be used on its own as a simple estimator of future phosphorus load, (e.g. as in Ockenden et al.¹⁵), but the models used here improve on this simple estimation by including the non-stationarity of the relationship which results from the change in rainfall distribution in the future and the resulting change in effective rainfall.”

l. 176-178: This seems contrary to the previous statements that even high reductions in P inputs are not able to mitigate the increase due to climate change. Perhaps the focus of the mitigation measures should be not only on P inputs but also other aspect of hydrological cycle, erosion, and sediment transport.

We did not mean to imply that other parts of the hydrological cycle should not be addressed too; indeed, we agree that they should be. We have clarified that we have used P inputs as an example (which models can handle transparently) which is representative of a range of different management options. We have added (p12 L264):

“The example of P inputs has been used here to demonstrate the relative scale of climate change and land management change impacts. We have modelled changing P inputs, which can be easily interpreted, to represent many more spatially specific mitigation measures, which models are not capable of representing without big increases in uncertainty. We are not suggesting that P inputs should be the sole focus of mitigation measures, indeed we recognise that such measures need to be catchment specific, addressing sources, mobilisation and transfer along the transfer continuum²⁹. There are many motives and challenges for farmers and stakeholders in choosing³⁰ or accepting³¹ mitigation options. Stakeholders in this study are already quite well focussed on appropriate measures for their specific catchments, but we demonstrate that these measures may not be enough in the face of the climate challenge.”

Reviewer #3 (Remarks to the Author):

“Major agricultural changes required to mitigate phosphorus losses under climate change” by Ockenden et al., submitted to Nature Communications

Review by Hoshin V Gupta, Professor, University of Arizona

Summary of the Main Message: This communication reports on the results of a modeling study, based on stakeholder inputs, to evaluate the impact of projected (2050 and beyond) climate change on land to water Phosphorus losses at three agricultural catchments in the UK, and to assess what scale of agricultural change would be needed to mitigate these transfers. The results suggest that average winter P loads can be expected to increase up to 30% by the 2050s and that large-scale agricultural changes resulting in 20-80% reduction in P inputs will be needed to limit the projected impacts, which is greater than considered technically feasible by use of farmer-preferred mitigation measures. The study recommends a) the prioritization of mitigation measures that reduce stores of P (and other pollutants) in the soil, and b) reduction in point sources that contribute to P loads. Because ‘legacy’ stores of ‘P (and nitrogen) have accumulated over decades, it may take several decades of reduced P input before the stores are depleted to mid-20th Century levels. This has implications for agricultural production, and may not be compatible with the need for increased agricultural productivity, requiring reassessment of priorities.

Reviewer Comments:

- 1) **Overall:** Overall, the study seems well conceived and executed, and the main message is clear (although could be enhanced as indicated below).
- 2) **Presentation:** I found the report a little difficult to read due to the manner in which the material was presented (i.e., organization), which also suffers from some degree of repetition. To gain a clear understanding, I found it useful to rearrange the presentation in the manner provided below (with some details omitted). I include this for the authors benefit, along with my summary above, in case they find it helpful in revising the manuscript to improve the presentation of their message and thereby increase its impact.

Thank you for the precis of the manuscript. We have used it to help revise and structure the presentation. We have added subheadings in Methods and Results to help organisation. We have reordered the results section as suggested, with modelling of the baseline conditions (relationship between annual rainfall and phosphorus load) before the climate change projections. We have not followed Professor Gupta’s order of the sections as this is different from that dictated by the journal (method section at end, discussion section rather than conclusions)

Subheadings in results are now:

Current phosphorus pollution sources and mitigation

Streamflow and phosphorus loads under current conditions

Projections under climate change only

Projections under combined climate and agricultural change

3) **Hydrological Modeling:** I was asked to specifically assess the hydrological modeling aspect of the study. Given the brief nature of this communication, very little actual modeling detail is provided and therefore I am not able to say very much substantive in this regard.

Further information is added in the Supplementary Information regarding model fit statistics: for HYPE, SI Table S4; for DBM, SI Table S6 and accompanying manuscript (to be submitted elsewhere) describing the development of the DBM phosphorus load model.

Overall, the fact that both a semi-distributed process-based daily time step model and a hourly time step databased mechanistic model give similar results suggests that the modeling results are probably reasonable, but this may also be a result of the fact that there is a clear and extremely strong relationship between rainfall and P load (Figure 3), at the annual level, and so the results likely do not depend strongly on the choice of hydrological model (except in degree). It might be interesting, therefore, for the authors to compare their results to a back-of-the envelope calculation based on only the explicit relationship shown in Figure 3, and to comment on any differences that might show up between the detailed modeling approach and a crude ‘quick and dirty’ assessment.

We have looked at the simple assessment of phosphorus load based on a linear relationship between (annual) rainfall and phosphorus load. This would be exactly the same as using a linear transfer function model, with rainfall as input and phosphorus load as output, which would result in the future changes in TP load being exactly the same as the future changes in rainfall. This can actually be seen in SI Table S1, where, for the Blackwater catchment, a linear DBM model between rainfall and TP load was used directly (because the non-linear representation of rainfall proved unstable in this catchment). All the future projections of changes in annual phosphorus load for the DBM model are identical to the future changes in annual rainfall. Seasonal changes are similar but not quite identical because of the effect of antecedent conditions in the transfer function model. In contrast, for the other catchments, or for the Blackwater catchment with HYPE, changes in annual and winter phosphorus load are predicted to be much higher than the corresponding changes in rainfall, due to the disproportional effect that high rainfall events have on phosphorus transfers. With both HYPE and DBM, we have used models which could include the non-linear effect that catchment storage has on the hydrological response, as this represents an improvement in modelling phosphorus transfers. We have added a comment (p11 L233):

“The dominance of high rainfall as a driver of phosphorus load (Figure 1) has been noted in other studies¹⁴, particularly during high erodibility periods. The relationship could be used on its own as a simple estimator of future phosphorus load, (e.g. as in Ockenden et al.¹⁵), but the models used here improve on this simple estimation by

including the non-stationarity of the relationship which results from the change in rainfall distribution in the future and the resulting change in effective rainfall.”

We have replotted Figure 1 (was Fig 3) to show rainfall against phosphorus load for every year of every run of the HYPE model, to show the variability in the relationship over the years. For each individual year the relationship gives a slightly higher regression coefficient than all years together, but the relationship for all years together is more appropriate if one wanted to make a linear estimate of phosphorus load for a given rainfall.

4) **Climate Modeling:** Similarly, it might be useful to clearly remark on whether there was any significant difference in assessed results using the new high-resolution convection-permitting regional climate model over the other approaches tested (I gather not, based on comments in the text). That is not to say that the convection-permitting model results are not useful, since they add confidence/credibility to the overall results.

It is difficult to interpret differences between results from the single run of high-resolution convection-permitting climate model and the probabilistic results from the UKCP Weather Generator because they cover different time spans (RCM-1.5km was run for 2100, whereas UKCP-WG was run for 2050 and 2080). In addition, from the single run from RCM-1.5km it is not possible to say where this would lie in a distribution. There are papers on the improved rainfall predictions from this model (Kendon et al., 2014; Chan et al., 2014; Kendon et al., 2017) which we have referenced. However, we have added a comment on the consistency of the results (p7 L147):

“Although the results from RCM-1.5km are not directly comparable to those using UKCP09-WG, because of the different time frame and the lack of uncertainty, they do not appear to show significant differences. However, the use of extra climate model, giving results that are consistent with those from UKCP09-WG adds further credence to the results.”

5) Such remarks (re hydrological and climate modeling) are useful in generally evaluating the broad policy-relevant benefits of very detailed modeling approaches (as opposed to evaluating specific local impacts and remediation strategies.

6) Towards the end the communication alludes to the “*recommendations of Michalak*” without clarifying for the general reader what those recommendations actually were/are. It would therefore be useful to do so.

Agreed. We have changed the comment (was L67, now p4, L68) to:

“Michalak notes that climate research and water quality research are usually conducted entirely separately, partly due to the often differing scales of interest, and recommends that for better understanding of climate change effects, we need to bring together the two disciplines. Our integrated, multi-disciplinary study follows these

recommendations, with the potential to contribute to the understanding of likely future P losses.”

7) It would be useful to clarify that the main climate change mechanism resulting in the P load changes is specifically changes in seasonal rainfall and not (if I understood correctly) other factors such as temperature etc. Further, please clarify whether it is simply increased amounts of rainfall or changes in rainfall intensities that is the controlling factor, or both. The term “climate change” is so general as to be almost non-informative when trying to understand how to respond in setting policy and mitigation strategies, so a little more specificity could be very helpful.

We have added (p8 L153):

“As the estimations of change in phosphorus load from the DBM and HYPE models are similar, this suggests that the main mechanism driving the changes in phosphorus load is the change in seasonal rainfall totals. Other factors, such as temperature, which are included in HYPE but not in the DBM model, may also contribute, but this contribution is small compared to the dominant driver. Similarly, the lack of significant difference between results using the convection-permitting climate model and UKCP09-WG indicate that although rainfall intensity may also be a contributing factor, it is not as important as the change in rainfall volumes.”

We have used Professor Gupta’s summary (below) to help restructure the revision, as detailed above.

SUMMARY OF MAIN POINTS IN THE PRESENTATION

The Problem

- 1) Climate change and intensification of agricultural food production pose threats to water quality and aquatic ecosystem functions and services.
- 2) Biogeochemical flows (specifically P) are already considered to be beyond the safe operating space for sustainable human development, and nutrient abatement strategies are needed.
- 3) Predicting future nutrient transfers into rivers, lakes and groundwater is challenging, due to the complexity of the landscape processes involved and the uncertainties in the input data, model structures and calibration data.
- 4) Previous studies have been limited by inadequate data resolution, lack of appropriate P data, limited model comparison, and lack of uncertainty analysis.

Study Goals

- 1) To use a combination of modeling and stakeholder-based inputs to evaluate the impact of projected climate change (to 2050 and beyond) on land to water Phosphorus (P) losses at three agricultural catchments in the UK
- 2) To assess what scale of agricultural change would be needed to mitigate these transfers.
- 3) To overcome the aforementioned limitations of previous studies.

Approach

To combine:

- 1) P flux data from three representative catchments across the UK
- 2) Projections provided by a new high spatial resolution climate model
- 3) Uncertainty estimates provided by an ensemble of future climate simulations
- 4) P transfer models of contrasting complexity
- 5) A simplified representation of the potential intensification of agriculture based on expert elicitation from land managers.

Methods

- 1) Models were developed using meteorological, flow and nutrient data available at hourly resolution for Oct 2011 – Sept 2014 (three years).
- 2) Two hydrological models of different complexity were applied; a) the process-based, semi-distributed water quality model HYPE run at a daily time step, and b) a databased continuous-time transfer function mechanistic model run at hourly time step. Model parameters were identified using the first two years and evaluated using the third year.
- 3) Both models were used to make future projections of discharge and P load under climate change. Future climatic rainfall and meteorological data were taken from three sources; a) a convection-permitting regional climate model (RCM-1.5km), b) the Met Office Unified Model, and c) data from the UKCP09 Weather Generator. For the UKCP09-WG, a 30 year probabilistic ensemble (100 runs) was generated for both baseline and scenario conditions.
- 4) Average annual loads were calculated using 26 years, with first four years as spin-up. For HYPE, results were averaged over all behavioral parameter sets. Winter loads were computed for Dec/Jan/Feb, and summer loads for Jun/Jul/Aug. Uncertainty estimates are based on the spread over 100 runs. For RCM-1.5km only 12-year simulations were used.
- 5) Future agricultural changes were determined via expert elicitation with stakeholders in each catchment. And used to guide modeling of future changes.
- 6) Future projections of P load were made under combined climate change and agricultural change scenarios using HYPE, by increasing or decreasing fertilizer and manure inputs, with no changes to point source inputs.

Results

Current conditions:

- 1) There is a clear relationship between annual rainfall and annual P load (Figure 3), illustrating the dominant effect of rainfall in driving diffuse P loads from agricultural lands

Considering climate change alone (2050s, high emissions scenario):

- 2) Pronounced seasonal changes in future rainfalls are predicted for the 2050s high emissions scenario (Table 1).
 - a. 14-15% increase in median winter rainfall
 - b. 14-19% reduction in summer rainfall
 - c. The higher resolution climate model shows even larger percentage changes
- 3) Inter-annual variability is very large, but both models predict very similar trends for future

P exports (Figure 1).

- a. Increased winter rainfall resulting in larger median winter flows and correspondingly larger (up to 31% increase) winter P loads.
- b. While summer decreases in median flow result in a (6-21%) decrease in median P load, the summer contribution to the annual load is small (typically less than 15%)

Considering climate change combined with future land management scenarios:

- 4) In the arable catchment, stakeholders favored the use of cover crops, while in the livestock-dominated catchments they favored increase in winter housing for livestock and increase in slurry spreading.
- 5) Projections of P load (Figure 2) indicate that reductions in P inputs would be required to offset the increase resulting from climate change.
- 6) Much larger P input reductions are required in the Avon, reflecting the different sources of P, which are not all agricultural (high background concentrations, sewage treatment works, rural septic tanks, apatite nodules in the underlying Chalk).

Conclusions

For the studied catchments:

- 1) The integrated approach used here supports the recommendations of Michalak and contributes to the understanding of likely future P losses.
- 2) Land to water P losses will be impacted by climate change and land management for food production, with detrimental impacts on aquatic ecosystems.
- 3) Climate change is predicted to increase average winter P loads up to 30% by 2050s,
- 4) Large-scale agricultural changes that result in 20-80% reduction in P inputs will be needed to limit the projected impacts of climate change on P loads.
- 5) This is greater than the technically feasible reduction (around 15% for catchment scale) from farmer-preferred mitigation measures estimated by previous studies.
- 6) Given the importance of rainfall for P transfers (particularly in winter when catchments are generally more saturated), it may be prudent for policy advisors and land managers to prioritize mitigation measures that reduce stores of P (and other pollutants) in the soil and address winter runoff. This could also lessen the impact of extreme events on in-stream ecological communities, which can be reset by large geomorphological changes and nutrient transfers that may occur during flooding.
- 7) In summer, unless point sources that contribute to P loads are reduced, reduced discharge could result in increased P concentrations and greater risk of ecological degradation.
- 8) 'Legacy' stores of 'P (and nitrogen) accumulated over decades present a challenge to water quality remediation. It may take several decades of reduced P input before the stores are depleted to levels last seen in the mid-20th Century. This could have implications for agricultural production, and may not be compatible with the need for increased agricultural productivity, requiring reassessment of priorities.
- 9) Due to the response lag times, and non-linear interactions between climate and agro-ecosystems, it is important to adopt an integrated approach to understanding climate effects on sustainable agriculture.
- 10) These findings are applicable to other agricultural regions in the world with temperate

climates where wetter winters are projected.

Table S4 HYPE model fit statistics (including seasonal statistics)

This table of model fit statistics includes the observed and modelled median (Q50 or TP50), and ranges (Q10, Q90 or TP10, TP90) for discharge Q and total phosphorus load TP. Nash Sutcliffe Efficiency (NSE) and model bias (PBIAS) are given for the calibration period (1 October 2011 – 30 September 2013, denoted ‘all’) and the validation period (1 October 2013 – 30 September 2014, denoted ‘all’) and for the winter season (December, January, February (DJF)) and the summer season (June, July, August (JJA)) within each of those periods, where data is available. Because the models were calibrated on the whole of the calibration period (and not specifically conditioned for winter or summer periods) the winter and summer model fit statistics may be misleading. This also reflects the shortcomings of NSE and PBIAS as model fit statistics – because NSE is based on the square of model residuals, it tends to be dominated by how well the model fits the peaks, which, for both discharge and total phosphorus load, tend to be higher and more frequent in winter. This frequently leads to overestimation in summer, where a (relatively) small absolute error on a very small quantity results in a very large percentage error (and hence misleading PBIAS). For this reason, this table should be read in conjunction with SI Figures S3-S14, which show the observed and modelled data for specified ‘best’ and ‘worst’ behavioural runs, where ‘best’ and ‘worst’ were subjectively chosen from the table below to illustrate the fit statistics in the context of the whole of the fit period.

Catchment	Newby Beck		Discharge Q (m ³ s ⁻¹)																	
	Calibration		0.0 % data missing			Validation			0.0 % data missing			PBIAS %			PBIAS %			Fig		
	Q10	Q50	Q90	NSE	PBIAS %			Q10	Q50	Q90	NSE	PBIAS %			Fig					
			all	DJF	JJA	all	DJF	JJA				all	DJF	JJA	all	DJF	JJA			
Observations	0.033	0.108	0.510								0.023	0.135	1.051							
Model	0.025	0.133	0.634	0.60	0.61	0.13	9.7	2.8	71.7		0.022	0.139	1.030	0.72	0.59	-1.80	-1.9	-9.5	129.0	
n = 14	0.033	0.114	0.537	0.61	0.62	0.45	-5.6	-1.7	26.1	S3	0.031	0.126	0.920	0.72	0.58	0.48	-12.4	-14.4	33.0	S4
	0.041	0.149	0.557	0.64	0.63	0.25	5.4	-3.8	65.5		0.038	0.165	0.891	0.72	0.57	-0.67	-5.9	-15.8	118.5	
	0.030	0.129	0.614	0.60	0.61	0.25	3.3	-6.2	62.6		0.030	0.138	0.962	0.70	0.54	-0.84	-8.1	-15.5	109.2	
	0.033	0.151	0.648	0.61	0.61	0.14	13.7	3.0	79.7		0.026	0.171	1.012	0.71	0.57	-1.97	0.7	-9.2	148.4	
	0.042	0.149	0.576	0.61	0.61	0.22	9.1	0.9	70.1		0.039	0.163	0.959	0.71	0.56	-1.10	-2.4	-11.7	127.6	
	0.067	0.150	0.515	0.61	0.60	0.42	3.7	4.4	44.4		0.059	0.153	0.942	0.71	0.55	0.34	-5.3	-13.2	75.4	
	0.066	0.188	0.610	0.60	0.61	0.12	21.3	0.5	104.5	S3	0.062	0.206	0.963	0.71	0.57	-2.73	4.7	-13.3	205.6	S4
	0.034	0.141	0.591	0.61	0.61	0.35	6.3	3.0	58.5		0.028	0.149	0.971	0.72	0.56	-0.19	-5.1	-12.4	92.4	
	0.031	0.104	0.493	0.60	0.59	0.41	-12.5	-10.4	23.0		0.030	0.105	0.879	0.69	0.51	0.56	-18.7	-20.1	37.5	
	0.023	0.147	0.635	0.61	0.60	0.21	10.1	3.8	67.8		0.016	0.157	1.004	0.71	0.56	-1.21	-1.9	-9.8	117.7	
	0.045	0.116	0.616	0.61	0.62	0.37	1.5	-2.7	46.8		0.048	0.129	0.951	0.69	0.52	-0.15	-8.6	-14.8	91.0	
	0.077	0.185	0.607	0.60	0.61	0.21	18.8	4.7	86.1		0.079	0.203	0.960	0.72	0.59	-1.00	4.2	-10.9	164.9	
	0.051	0.136	0.510	0.60	0.59	0.36	-2.1	-3.7	44.1		0.049	0.144	0.873	0.70	0.53	0.26	-11.4	-18.4	80.0	

Table S4 contd. HYPE model fit statistics

Catchment	Newby Beck		Total phosphorus load TP (kg day ⁻¹)							Fig	Validation		% data missing			PBIAS %			Fig	
	Calibration		17.2		% data missing						34.5		% data missing			PBIAS %				
	TP10	TP50	TP90	NSE			PBIAS %				TP10	TP50	TP90	NSE			PBIAS %			
				all	DJF	JJA	all	DJF	JJA					all	DJF	JJA	all	DJF		JJA
Observations	0.116	0.509	8.610								0.026	0.743	18.776							
Model	0.053	0.558	6.802	0.44	0.55	0.24	-43.0	-33.4	-36.3		0.034	0.512	14.556	0.47	0.33	-51.16	-33.4	-42.1	797.8	
n=14	0.112	0.716	11.162	0.50	0.49	0.03	15.7	39.9	20.6		0.121	0.748	26.387	0.45	0.33	-81.49	35.4	27.6	1083.7	
	0.135	1.103	9.242	0.60	0.71	0.32	-18.6	-11.4	0.0		0.112	1.074	19.808	0.59	0.53	-107.69	-7.4	-21.4	1171.8	
	0.084	0.543	4.433	0.48	0.52	0.23	-47.2	-45.7	-36.3		0.083	0.568	11.111	0.44	0.28	-36.61	-43.1	-50.3	690.4	
	0.234	1.553	13.844	0.62	0.69	0.24	19.0	33.4	41.6		0.158	1.539	27.331	0.59	0.59	-210.34	31.2	11.7	1729.0	
	0.094	0.662	8.250	0.59	0.67	0.24	-19.5	-10.0	-7.8		0.087	0.661	19.764	0.57	0.50	-120.57	-6.5	-18.2	1204.5	
	0.155	0.770	8.426	0.60	0.69	0.26	-15.2	7.8	-10.3		0.172	0.859	20.166	0.64	0.55	-32.01	3.7	-4.6	814.8	S6
	0.222	1.248	9.313	0.54	0.66	0.30	-19.5	-16.1	3.8		0.202	1.379	19.144	0.56	0.49	-105.09	-8.4	-25.0	1346.1	
	0.123	0.956	9.522	0.60	0.72	0.33	-15.6	-0.1	-5.2		0.103	0.978	20.524	0.64	0.56	-58.46	-5.1	-16.6	1017.0	
	0.066	0.762	11.714	0.62	0.65	0.30	2.1	32.0	2.1	S5	0.091	0.777	26.016	0.63	0.53	-22.68	22.1	16.1	675.3	
	0.186	1.878	17.442	0.56	0.57	0.09	44.7	62.5	73.9	S5	0.095	1.784	31.768	0.47	0.53	-365.17	54.9	30.8	2160.8	S6
	0.057	0.295	5.502	0.52	0.61	0.17	-38.8	-39.2	-22.1		0.073	0.336	15.306	0.46	0.34	-86.40	-33.9	-42.0	950.8	
	0.224	0.876	6.160	0.49	0.60	0.23	-35.0	-23.0	-28.3		0.238	0.974	15.142	0.57	0.45	-24.38	-20.4	-29.4	723.1	
	0.177	1.066	10.742	0.57	0.65	0.28	-2.6	28.6	-3.1		0.185	1.106	23.188	0.59	0.49	-41.09	17.4	7.2	879.7	

Table S4 contd. HYPE model fit statistics

Catchment	Blackwater		Discharge Q (m ³ s ⁻¹)							Fig	Validation		% data missing			PBIAS %			Fig	
	Calibration		14.9		NSE			PBIAS %			1.9		NSE			PBIAS %				
	Q10	Q50	Q90	NSE	all	DJF	JJA	all	DJF		JJA	Q10	Q50	Q90	NSE	all	DJF	JJA		all
Observations	0.025	0.062	0.236								0.025	0.067	0.210							
Model	0.013	0.048	0.251	0.57	0.82	-2.95	-10.4	-15.9	40.9		0.016	0.041	0.162	-0.18	-0.67	-19.69	-21.7	-42.0	58.2	S8
n=12	0.013	0.046	0.260	0.57	0.77	-2.83	-10.8	-18.9	39.3		0.017	0.040	0.170	-0.32	-0.74	-17.01	-18.6	-40.7	61.1	
	0.011	0.046	0.249	0.55	0.75	-3.27	-13.6	-18.2	37.2		0.014	0.042	0.154	-0.22	-0.76	-17.05	-24.4	-45.6	60.8	
	0.012	0.048	0.254	0.56	0.79	-2.82	-11.6	-17.8	37.8		0.016	0.043	0.175	-0.21	-0.65	-17.96	-20.7	-41.7	65.6	
	0.012	0.045	0.244	0.55	0.79	-3.87	-12.5	-20.9	42.4		0.014	0.039	0.163	-0.39	-0.69	-18.15	-22.7	-45.3	54.4	
	0.013	0.053	0.263	0.57	0.80	-3.70	-3.4	-9.6	53.4		0.017	0.048	0.181	-0.28	-0.51	-20.80	-13.3	-38.3	83.5	
	0.011	0.042	0.236	0.56	0.77	-1.56	-19.9	-22.1	13.8	S7	0.014	0.037	0.145	-0.19	-0.86	-16.87	-28.4	-46.0	49.4	
	0.011	0.049	0.262	0.55	0.77	-3.89	-9.6	-18.6	48.4		0.015	0.044	0.181	-0.38	-0.55	-19.77	-17.6	-44.2	75.4	
	0.010	0.044	0.248	0.56	0.79	-3.85	-11.3	-21.6	39.5		0.013	0.038	0.182	-0.68	-0.62	-17.52	-18.7	-43.2	58.1	S8
	0.013	0.048	0.236	0.57	0.76	-2.85	-13.8	-20.4	37.2		0.015	0.042	0.153	-0.26	-0.64	-18.64	-23.1	-46.3	61.8	
	0.011	0.051	0.264	0.55	0.82	-3.95	-7.2	-14.9	49.2	S7	0.014	0.043	0.176	-0.42	-0.61	-22.56	-15.8	-40.1	76.4	
	0.012	0.051	0.249	0.56	0.78	-2.67	-10.2	-14.8	40.2		0.015	0.043	0.166	-0.34	-0.70	-17.86	-20.8	-45.4	66.1	

Table S4 contd. HYPE model fit statistics

Catchment	Blackwater		Total phosphorus load TP (kg day ⁻¹)							Fig	Validation			% data missing			Fig			
	Calibration		40.2								12.3									
	TP10	TP50	TP90	NSE			PBIAS %				TP10	TP50	TP90	NSE				PBIAS %		
				all	DJF	JJA	all	DJF	JJA					all	DJF	JJA		all	DJF	JJA
Observations	0.188	0.500	1.688								0.177	0.428	1.275							
Model	0.079	0.372	1.868	0.53	0.42	-0.28	-17.2	-25.5	7.6		0.153	0.328	1.019	-0.22	0.18	-39.56	-10.6	-54.3	99.1	
n=12	0.072	0.381	1.985	0.53	0.44	-0.21	-16.1	-22.5	6.5		0.155	0.317	0.972	-0.21	0.10	-28.44	-12.2	-50.9	91.3	
	0.080	0.456	1.883	0.53	0.44	-0.94	-12.4	-21.3	20.8		0.147	0.381	1.176	-0.37	0.16	-38.75	-3.8	-53.3	123.1	
	0.084	0.459	1.943	0.54	0.43	-0.76	-10.9	-22.2	17.6		0.168	0.378	1.176	-0.28	0.21	-37.68	-1.2	-47.3	115.2	
	0.070	0.419	1.970	0.55	0.47	-1.06	-12.4	-21.7	23.5		0.151	0.336	1.131	-0.30	0.21	-38.24	-6.0	-49.7	104.5	
	0.079	0.428	1.714	0.53	0.37	0.49	-17.2	-21.0	-11.1		0.145	0.338	1.091	0.06	0.15	-18.86	-14.4	-49.7	75.2	
	0.063	0.390	1.901	0.53	0.41	0.06	-18.0	-24.0	-3.8		0.140	0.317	1.007	-0.11	0.21	-30.74	-11.3	-48.3	91.4	
	0.065	0.408	1.625	0.53	0.37	0.00	-23.4	-29.8	-0.9	S9	0.137	0.328	1.043	-0.27	0.12	-32.74	-15.0	-60.9	94.8	
	0.061	0.368	1.640	0.54	0.36	0.31	-23.6	-31.6	-11.1		0.125	0.282	1.031	-0.13	0.11	-20.27	-20.3	-54.4	57.8	
	0.091	0.444	1.723	0.54	0.40	-0.40	-15.8	-22.9	5.1		0.153	0.352	1.116	-0.09	0.18	-32.20	-9.9	-50.9	97.7	
	0.063	0.437	1.910	0.53	0.44	-0.49	-12.6	-23.0	7.7		0.132	0.321	1.133	-0.58	0.20	-49.51	-2.9	-49.7	108.5	
	0.079	0.474	2.085	0.55	0.48	-0.09	-8.1	-12.7	10.2	S9	0.155	0.381	1.146	-0.22	0.21	-30.94	-1.5	-45.4	112.3	

Table S4 contd. HYPE model fit statistics

Catchment	Wylye		Discharge Q (m ³ s ⁻¹)							Fig	Validation			% data missing			Fig			
	Calibration		0.0	% data			PBIAS %				Q10	Q50	38.4	PBIAS %						
	Q10	Q50	Q90	NSE	all DJF JJA			all	DJF		JJA	Q90	NSE	all DJF JJA						
Observations	0.082	0.184	0.784								0.069	0.366	1.765							
Model	0.098	0.232	0.653	0.65	0.69	-0.41	-4.8	-16.1	32.3		0.189	0.472	1.885	0.53	0.38	NaN	21.6	13.0	NaN	S12
n=11	0.076	0.210	0.625	0.64	0.66	-0.11	-3.8	-9.0	24.8		0.160	0.452	2.271	0.39	0.22	NaN	29.0	26.8	NaN	
	0.082	0.192	0.680	0.61	0.70	-0.75	-5.0	-15.3	38.5	S11	0.157	0.453	2.195	0.28	0.02	NaN	29.0	29.8	NaN	S12
	0.044	0.131	0.535	0.60	0.61	0.00	-30.5	-30.7	-10.1		0.102	0.259	1.396	0.42	0.15	NaN	-14.8	-12.1	NaN	
	0.056	0.168	0.649	0.65	0.68	-0.20	-16.2	-18.7	15.1		0.156	0.364	1.969	0.52	0.37	NaN	13.9	17.9	NaN	
	0.070	0.197	0.686	0.66	0.70	-0.28	-9.0	-17.7	28.8		0.156	0.487	2.089	0.27	-0.14	NaN	28.8	27.7	NaN	
	0.075	0.163	0.560	0.64	0.67	-0.23	-19.6	-29.2	17.7		0.147	0.364	1.525	0.43	0.20	NaN	-4.0	-7.5	NaN	
	0.033	0.135	0.690	0.62	0.72	-0.74	-17.2	-19.7	16.8		0.109	0.365	2.194	0.52	0.29	NaN	19.4	24.2	NaN	
	0.079	0.218	0.620	0.64	0.65	-0.30	-4.8	-12.1	31.0		0.180	0.420	2.262	0.37	0.10	NaN	28.6	29.5	NaN	
	0.053	0.153	0.654	0.62	0.69	-0.42	-19.1	-19.1	8.1		0.137	0.329	2.246	0.31	-0.07	NaN	21.0	27.5	NaN	
	0.128	0.250	0.645	0.65	0.66	-0.19	-0.5	-20.7	47.8	S11	0.206	0.547	1.740	0.43	0.27	NaN	19.7	6.5	NaN	

Table S4 contd. HYPE model fit statistics

Catchment	Wylfe		Total phosphorus load TP (kg day ⁻¹)							Fig	Validation			% data missing			Fig			
	Calibration		49.9		% data			67.7			Validation			% data missing						
	TP10	TP50	TP90	NSE			PBIAS %				TP10	TP50	TP90	NSE				PBIAS %		
				all	DJF	JJA	all	DJF	JJA					all	DJF	JJA		all	DJF	JJA
Observations	0.817	3.048	12.117								1.089	1.353	30.979							
Model	0.635	2.473	8.418	0.64	-0.15	0.61	-29.3	-36.9	-3.2		0.783	10.954	28.537	-0.07	-0.42	NaN	22.8	4.3	NaN	
n=11	0.628	3.014	12.428	0.62	0.14	-0.23	-8.4	-3.7	-2.3		1.641	14.443	43.730	-0.21	-0.49	NaN	67.6	40.2	NaN	
	0.516	2.580	10.951	0.63	0.07	-0.24	-15.7	-16.8	-20.4		1.136	14.317	45.954	-0.42	-0.89	NaN	72.9	47.6	NaN	
	0.475	2.245	10.118	0.66	0.10	-0.96	-21.1	-24.6	-35.1		0.333	9.822	29.753	-0.06	-0.46	NaN	25.1	9.3	NaN	
	0.565	2.864	11.661	0.65	0.09	0.52	-7.0	-0.8	-11.7		0.513	14.758	42.935	-0.69	-1.33	NaN	76.0	52.1	NaN	
	0.561	2.775	9.903	0.62	-0.02	0.49	-17.8	-25.3	-11.0		0.643	15.087	41.401	-0.87	-1.62	NaN	66.0	40.8	NaN	
	0.718	2.314	8.058	0.62	-0.22	0.60	-29.7	-38.6	0.7	S13	0.657	8.198	26.101	-0.11	-0.39	NaN	15.0	-2.0	NaN	
	0.554	2.409	12.201	0.65	0.04	-0.02	-8.1	-3.8	-21.0		0.461	8.628	43.431	0.03	-0.35	NaN	54.0	40.0	NaN	
	0.578	2.641	10.255	0.63	0.02	0.30	-19.5	-22.4	-16.7		0.706	11.841	34.036	0.25	0.15	NaN	31.7	10.6	NaN	S14
	0.429	2.324	10.452	0.68	-0.03	-1.21	-17.7	-17.9	-36.8	S13	0.339	11.013	46.752	-1.86	-3.39	NaN	74.8	58.7	NaN	S14
	0.886	2.979	9.733	0.63	0.00	-0.80	-16.2	-22.3	30.2		1.649	13.377	37.148	-0.70	-1.25	NaN	51.9	27.3	NaN	

Table S6 DBM Model fit statistics

This table of model fit statistics includes the observed and modelled median (Q50 or TP50), and ranges (Q10, Q90 or TP10, TP90) for discharge Q and total phosphorus load TP. Nash Sutcliffe Efficiency (NSE) and model bias (PBIAS) are given for the calibration period (1 October 2011 – 30 September 2013, denoted ‘all’) and the validation period (1 October 2013 – 30 September 2014, denoted ‘all’) and for the winter season (December, January, February (DJF)) and the summer season (June, July, August (JJA)) within each of those periods, where data is available.

Catchment	Newby Beck		Runoff Q (mm h ⁻¹)							Validation		PBIAS %						
	Calibration		0.0	% data missing						0.0	% data missing							
	Q10	Q50	Q90	NSE			PBIAS %			Q10	Q50	Q90	NSE			PBIAS %		
				all	DJF	JJA	all	DJF	JJA				all	DJF	JJA	all	DJF	JJA
Observations	0.017	0.047	0.159							0.012	0.060	0.261						
Model	0.006	0.037	0.175	0.71	0.83	0.27	-9.7	-26.8	45.0	0.008	0.047	0.249	0.78	0.83	-2.04	-14.3	-21.4	95.1

Catchment	Newby Beck		Total phosphorus load TP (kg h ⁻¹)							Validation		PBIAS %						
	Calibration		13.2	% data missing						30.8	% data missing							
	TP10	TP50	TP90	NSE			PBIAS %			TP10	TP50	TP90	NSE			PBIAS %		
				all	DJF	JJA	all	DJF	JJA				all	DJF	JJA	all	DJF	JJA
Observations	0.009	0.032	0.267							0.005	0.048	0.490						
Model	0.000	0.001	0.474	0.65	0.62	0.53	2.3	-12.3	23.1	0.000	0.002	0.858	0.62	0.72	19.58	5.1	-13.1	399.2

Table S6 contd. DBM Model fit statistics

Catchment	Blackwater			Runoff Q (mm h ⁻¹)						Validation			PBIAS %					
	Calibration			4.8			% data missing			2.8			% data missing			PBIAS %		
	Q10	Q50	Q90	NSE			PBIAS %			Q10	Q50	Q90	NSE			PBIAS %		
			all	DJF	JJA	all	DJF	JJA				all	DJF	JJA	all	DJF	JJA	
Observations	0.005	0.011	0.044							0.005	0.012	0.038						
Model	0.004	0.016	0.039	0.37	0.72	-6.10	-1.5	-36.8	159.4	0.005	0.014	0.029	0.32	0.55	-2.97	-9.4	-51.7	176.7

Catchment	Blackwater			Total phosphorus load TP (kg h ⁻¹)						Validation			PBIAS %					
	Calibration			13.6			% data missing			16.8			% data missing			PBIAS %		
	TP10	TP50	TP90	NSE			PBIAS %			TP10	TP50	TP90	NSE			PBIAS %		
			all	DJF	JJA	all	DJF	JJA				all	DJF	JJA	all	DJF	JJA	
Observations	0.010	0.031	0.082							0.017	0.028	0.076						
Model	0.009	0.031	0.117	0.62	0.72	17.26	5.4	-5.2	62.9	0.017	0.048	0.130	0.03	0.48	43.34	38.2	-1.1	387.3

Table S6 contd. DBM Model fit statistics

Catchment	Wylie			Runoff Q (mm h ⁻¹)						Validation			PBIAS %					
	Calibration			0.3	% data missing			1.0			% data missing							
	Q10	Q50	Q90	NSE	PBIAS %			Q10	Q50	Q90	NSE	PBIAS %						
			all	DJF	JJA	all	DJF	JJA				all	DJF	JJA	all	DJF	JJA	
Observations	0.004	0.020	0.075							0.006	0.091	0.140						
Model	0.004	0.019	0.078	0.87	0.79	-0.35	3.0	0.1	-4.3	0.003	0.108	0.159	0.79	0.79	NaN	10.9	9.3	NaN

Catchment	Wylie			Total phosphorus load TP (kg h ⁻¹)						Validation			PBIAS %					
	Calibration			27.2	% data missing			46.6			% data missing							
	TP10	TP50	TP90	NSE	PBIAS %			TP10	TP50	TP90	NSE	PBIAS %						
			all	DJF	JJA	all	DJF	JJA				all	DJF	JJA	all	DJF	JJA	
Observations	0.036	0.167	0.543							0.055	0.876	1.615						
Model	0.041	0.181	0.625	0.55	0.30	16.64	5.5	4.5	30.8	0.001	0.688	1.550	0.50	0.52	NaN	-19.7	-22.6	NaN

Figure S3 Examples of behavioural runs for HYPE model calibration for Newby Beck discharge. Best behavioural run, $i = 2$ out of 14 (top), worst behavioural run, $i = 8$ out of 14 (bottom). Observed = blue; Modelled = red. For model fit statistics, see Table S4

Figure S4 Examples of behavioural runs for HYPE model validation for Newby Beck discharge. Best behavioural run, $i = 2$ out of 14 (top), worst behavioural run, $i = 8$ out of 14 (bottom). Observed = blue; Modelled = red. For model fit statistics, see Table S4

Figure S5 Examples of behavioural runs for HYPE model calibration for Newby Beck total phosphorus load (TP). Best behavioural run, $i = 10$ out of 14 (top), worst behavioural run, $i = 11$ out of 14 (bottom). Observed = blue; Modelled = red. For model fit statistics, see Table S4

Figure S6 Examples of behavioural runs for HYPE model validation for Newby Beck total phosphorus load. Best behavioural run, $i = 7$ out of 14 (top), worst behavioural run, $i = 11$ out of 14 (bottom). Observed = blue; Modelled = red. For model fit statistics, see Table S4

Figure S7 Examples of behavioural runs for HYPE model calibration for Blackwater discharge. Best behavioural run, $i = 7$ out of 12 (top), worst behavioural run, $i = 11$ out of 12 (bottom). Observed = blue; Modelled = red. For model fit statistics, see Table S4

Figure S8 Examples of behavioural runs for HYPE model validation for Blackwater discharge. Best behavioural run, $i = 1$ out of 12 (top), worst behavioural run, $i = 9$ out of 12 (bottom). Observed = blue; Modelled = red. For model fit statistics, see Table S4

Figure S9 Examples of behavioural runs for HYPE model calibration for Blackwater total phosphorus load. Best behavioural run, $i = 12$ out of 12 (top), worst behavioural run, $i = 8$ out of 12 (bottom). Observed = blue; Modelled = red. For model fit statistics, see Table S4

Figure S10 Examples of behavioural runs for HYPE model validation for Blackwater total phosphorus load. Best behavioural run, $i = 6$ out of 12 (top), worst behavioural run, $i = 11$ out of 12 (bottom). Observed = blue; Modelled = red. For model fit statistics, see Table S4

Figure S11 Examples of behavioural runs for HYPE model calibration for Wylve discharge. Best behavioural run, $i = 11$ out of 11 (top), worst behavioural run, $i = 3$ out of 11 (bottom). Observed = blue; Modelled = red. For model fit statistics, see Table S4

Figure S12 Examples of behavioural runs for HYPE model validation for Wylie discharge. Best behavioural run, $i = 1$ out of 11 (top), worst behavioural run, $i = 3$ out of 11 (bottom). Observed = blue; Modelled = red. For model fit statistics, see Table S4

Figure S13 Examples of behavioural runs for HYPE model calibration for Wylve total phosphorus load. Best behavioural run, $i = 10$ out of 11 (top), worst behavioural run, $i = 7$ out of 11 (bottom). Observed = blue; Modelled = red. For model fit statistics, see Table S4

Figure S14 Examples of behavioural runs for HYPE model validation for Wylle total phosphorus load. Best behavioural run, $i = 9$ out of 11 (top), worst behavioural run, $i = 10$ out of 11 (bottom). Observed = blue; Modelled = red. For model fit statistics, see Table S4

Reviewers' Comments:

Reviewer #1:

Remarks to the Author:

I enjoyed reading the revised manuscript as well as the very thoughtful responses to the reviewer's comments (mine included). Even small revisions have helped with the clarity. I have very little to comment upon at this point.

Reviewer #2:

Remarks to the Author:

The authors have significantly revised the manuscript which has increased the quality of the publication and the information contained there. All my comments have been addressed adequately.

There are several places where a minor editing is needed after the revision.

Reviewer #3:

Remarks to the Author:

Thanks for responding to my comments. I have no further remarks to make.

Response to reviewers NCOMMS-16-28846A

REVIEWERS' COMMENTS:

Reviewer #1 (Remarks to the Author):

I enjoyed reading the revised manuscript as well as the very thoughtful responses to the reviewer's comments (mine included). Even small revisions have helped with the clarity. I have very little to comment upon at this point.

Thank you

Reviewer #2 (Remarks to the Author):

The authors have significantly revised the manuscript which has increased the quality of the publication and the information contained there. All my comments have been addressed adequately.

There are several places where a minor editing is needed after the revision.

No specific edits were noted, but some minor changes have been made following the editorial revisions – all are marked in the tracked changes copy of the manuscript.

Reviewer #3 (Remarks to the Author):

Thanks for responding to my comments. I have no further remarks to make.

Thank you.